# VSTAR: Generative Temporal Nursing for Longer Dynamic Video Synthesis

**Yumeng Li**[1][*] **William Beluch**[2] **Margret Keuper**[3,4] **Dan Zhang**[2] **Anna Khoreva**[5*]

[1]Amazon [2]Bosch Center for Artificial Intelligence [3]University of Mannheim
[4]Max Planck Institute for Informatics [5]Zalando

`yumengll@amazon.com, {william.beluch,dan.zhang2}@de.bosch.com`
`keuper@uni-mannheim.de, anna.khoreva@zalando.de`
Project page: https://yumengli007.github.io/VSTAR

## ABSTRACT

Despite tremendous progress in the field of text-to-video (T2V) synthesis, open-sourced T2V diffusion models struggle to generate longer videos with dynamically varying and evolving content. They tend to synthesize quasi-static videos, ignoring the necessary visual change-over-time implied in the text prompt. Meanwhile, scaling these models to enable longer, more dynamic video synthesis often remains computationally intractable. To tackle this challenge, we introduce the concept of Generative Temporal Nursing (GTN), where we adjust the generative process on the fly during inference to improve control over the temporal dynamics and enable generation of longer videos. We propose a method for GTN, dubbed VSTAR, which consists of two key ingredients: **V**ideo **S**ynopsis Prompting (VSP) and **T**emporal **A**ttention **R**egularization (TAR), the latter being our core contribution. Based on a systematic analysis, we discover that the temporal units in pretrained T2V models are crucial to control the video dynamics. Upon this finding, we propose a novel regularization technique to refine the temporal attention, enabling training-free longer video synthesis in a single inference pass. For prompts involving visual progression, we leverage LLMs to generate video synopsis - description of key visual states - based on the original prompt to provide better guidance along the temporal axis. We experimentally showcase the superiority of our method in synthesizing longer, visually appealing videos over open-sourced T2V models.

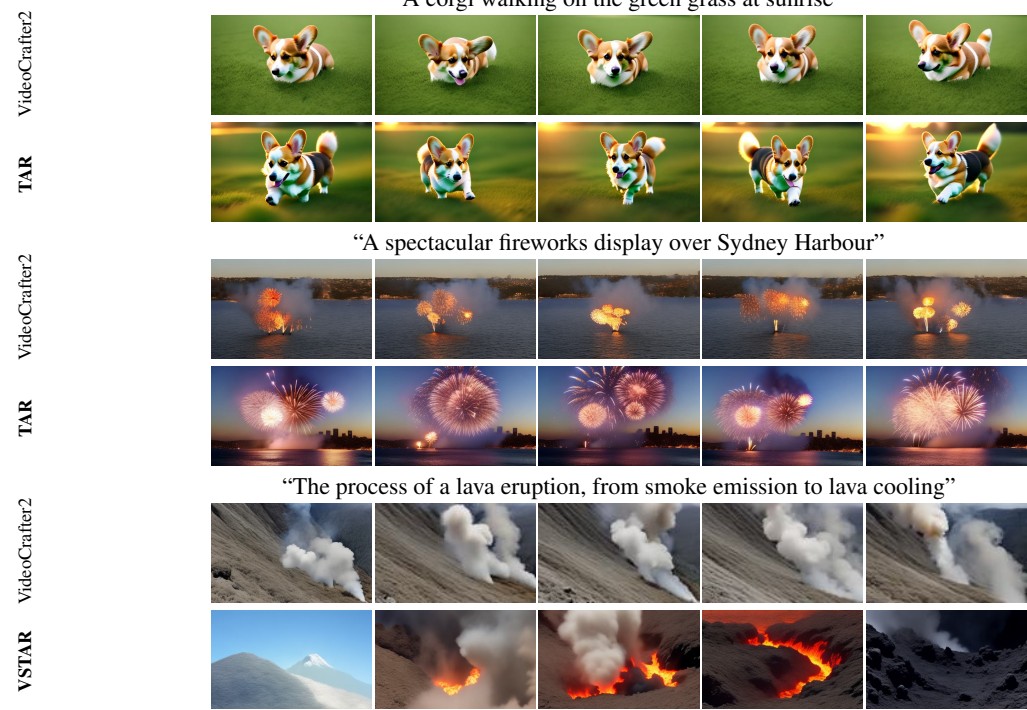

Figure 1: Our Temporal Attention Regularization (TAR) enables longer video generation, e.g., 128 frames, in a *single* pass. When the initial prompt involves progression with multiple stages (3rd row), enabled by Video Synopsis Prompting (VSP), VSTAR can synthesize dynamic visual evolution, i.e., from lava eruption, flowing, to cooling. The first column is a GIF, best viewed in *PDF Reader*.

---

[*]Work done at Bosch Center for Artificial Intelligence

# 1 INTRODUCTION

In recent years, rapid advances in the research and open-source community have led to significant progress in text-to-image synthesis, as well as its natural extension to text-to-video synthesis. Having transformed the idea of content creation, they are now widespread as both a research topic and an industry application. In the realm of text-to-video (T2V) synthesis specifically, recent advancements in video diffusion models (Blattmann et al., 2023; Wang et al., 2023c;b; Guo et al., 2024; Chen et al., 2024a; OpenAI, 2024) have offered promising possibilities for creating novel video content from textual descriptions.

However, despite these advancements, we observe two common issues in current open-source T2V models (Wang et al., 2023b;c; Guo et al., 2024; Chen et al., 2024a): limited visual changes within the video, and a poor ability to generate longer videos with coherent temporal dynamics. More specifically, the synthesized scenes often exhibit a high degree of similarity between frames (see Figs. 1, 7 and S.7), frequently resembling a static image with minor variations as opposed to a video with varying and evolving content. Additionally, these models do not generalize well to generate videos with more than the typical 16 frames in one pass (see Fig. 7). While several recent works attempt to generate long videos in a sliding window fashion (Wang et al., 2023a; Qiu et al., 2024), these methods not only introduce considerable overhead due to requiring multiple inference passes, but also face the new challenge of preserving temporal coherence throughout the passes.

To mitigate the aforementioned issues, and remain computationally affordable, we propose the training-free concept - *"Generative Temporal Nursing"* (GTN) - which aims to enhance the temporal dynamics of longer video synthesis in a single pass, on the fly during inference, without the need for retraining T2V models. As a form of GTN, we propose VSTAR, consisting of **V**ideo **S**ynopsis Prompting (VSP) and **T**emporal **A**ttention **R**egularization (TAR).

Most open-sourced T2V models, such as ModelScope (Wang et al., 2023b), LaVie (Wang et al., 2023c), and VideoCrafter (Chen et al., 2023; 2024a), are built upon T2I models, and process all frames within one batch. The single text prompt is conditioned via cross-attention in the spatial transformer of the UNet and shared by all frames. Different from T2I models with only spatial modules, the newly incorporated temporal attention units in T2V models serve as a critical component in driving the dynamic aspects of video synthesis, thus has significant impact on the outcome. Therefore, we systematically investigate these temporal units, which are based on temporal transformers consisting of self-attention layers. The visual gap between real videos and synthesized ones leads us to compare their temporal attention maps (see Fig. 3). We discover that real videos have a band-matrix-like structure, with strong correlation between adjacent frames and weaker correlation as distance increases. Intriguingly, the attention maps of the synthesized ones are less structured, explaining their inferior temporal dynamics.

Inspired by this observation, we propose a simple yet effective Temporal Attention Regularization (TAR) technique, which improves the video dynamics of generated videos and enables generation of longer videos (see Figs. 1 and 5). More specifically, we design a symmetric Toeplitz matrix with values along the off-diagonal direction following a Gaussian distribution. The standard deviation of this distribution can control the regularization strength, i.e., the visual variation along the temporal dimension. Adding it to the existing temporal attention maps strengthens the temporal correlation between adjacent frames, while reducing it between more distant frames. Notably, TAR is readily applicable to pre-trained T2V models and requires no optimization, thus introducing no extra inference overhead. Enabled by TAR, we additionally demonstrate a user-friendly way to transfer dynamics from a given reference video to the generated one, by employing its inter-frame similarity as the temporal regularization matrix (see Fig. 8).

When the user input describes scenarios that involve multi-stage changes or high dynamics, as shown in the 3rd row of Fig. 1, T2V models struggle to transform the semantics from a single prompt into the required visual change across frames. Prior T2I works (Betker et al., 2023; Yang et al., 2024a) have utilized large language models (LLMs) to decompose a single prompt containing multiple entities into several subprompts for spatial compositional generation. Analogously, for dynamic video synthesis faithful to the input prompt, the generation could benefit from a *synopsis* that describes the main events of the video, with explicit descriptions about the desired visual development over time. As a component of GTN, our Video Synopsis Prompting (VSP) leverages LLMs to generate key visual states, thus providing the T2V model more accurate guidance from the spatial perspective, i.e., via cross attention. To ensure temporal coherency, we propose to interpolate between textual embeddings

of key stages, ensuring smooth conditioning without abrupt changes between frames. Equipped with both strategies, our VSTAR can produce long videos with appealing visual changes in one single inference pass (see Figs. 7 and S.5).

Finally, in addition to improving longer video synthesis, we analyze the temporal attention mechanisms of different T2V models, establishing valuable connections between their capability to generate longer videos and their architectures. Following the analysis, we offer several training suggestions for enhancing the generalization ability of future models. In summary, our contributions include:

- We introduce a novel concept of "Generative Temporal Nursing", aiming to improve temporal dynamics, especially for long videos, without training needed or introducing high computing overhead at inference time.
- We propose VSTAR, a method for Generative Temporal Nursing, consisting of two simple yet effective strategies: Video Synopsis Prompting and Temporal Attention Regularization, which enable long video generation in a single inference pass with improved video dynamics.
- We are the first to provide an analysis of temporal attention within video diffusion models, and unleash its potential for controlling the video dynamics. Based on the analysis, we provide insights on how to improve the training of the next generation of T2V models.

## 2 RELATED WORK

**Text-to-Video Diffusion Models.** Remarkable progress in T2V has been made by industrial research labs (OpenAI, 2024; Kaishou, 2024; AI, 2024; Runway, 2024), however, their models are not publicly accessible. Therefore, we focus on investigating open-sourced T2V diffusion models (Blattmann et al., 2023; Wang et al., 2023b;c; Chen et al., 2023; Guo et al., 2024; Chen et al., 2024a), which are commonly built upon large-scale pretrained T2I model, e.g., Stable Diffusion (Rombach et al., 2022). Such methods generally introduce a temporal dimension to the T2I model and include temporal transformer for temporal modeling with finetuning on a video dataset. A few recent works, e.g., Open-Sora-Plan (Lab & etc., 2024) and CogVideoX (Yang et al., 2024b) have attempted to reproduce Sora using 3D full attention directly, without explicitly modeling the temporal dimension (i.e., without using temporal attention modules), which falls outside the scope of our approach.

Since long video generation is especially difficult, there are also works (Qiu et al., 2024; Wang et al., 2023a) focusing specifically on this application. FreeNoise (Qiu et al., 2024) proposed noise rescheduling combined with local window based attention fusion. Gen-L-Video (Wang et al., 2023a) casts the problem as fusing multiple short video clips with temporal overlapping. However, they necessitate several passes for generation, significantly raising the inference overhead. Some recent works (Kondratyuk et al., 2023; Long et al., 2024; Zhuang et al., 2024) incorporated large-scale training and aim at multi-scene storytelling, allowing significant scene discontinuity, with the final video being a concatenation of multiple runs. Different from these methods, our VSTAR targets coherent long video generation with a *pretrained* T2V model in one *single* pass.

**Attention Manipulation.** In the realm of T2I models, many works (Cao et al., 2023; Chefer et al., 2023; Li et al., 2023; Hertz et al., 2023; Chen et al., 2024b) have identified the attention layers as potential targets to manipulate for improving synthesis. Chefer et al. (2023); Li et al. (2023) employs inference time latent optimization based on the cross-attention maps, to enhance faithfulness to the input prompt, e.g., encourage object presence and proper attribute binding. However, such optimization increases the computation cost at inference time. There are also methods (Hertz et al., 2023; Feng et al., 2023) which directly modify or reweight the attention maps to enable text-controlled image editing or improve attribute binding and compositionality. Nonetheless, for T2V diffusion models, there is still a lack of comprehensive understanding of the temporal attention mechanism. Our work is the first to investigate this aspect and unleash its manipulation potential for improving video generation of pretrained T2V models without extra optimization overhead during inference.

## 3 METHOD

Our concept of Generative Temporal Nursing (GTN) aims at improving the video dynamics of pretrained T2V diffusion models. Besides the text prompt, we identify in Section 3.1 that the temporal attention layer is another key component of T2V models responsible for determining video dynamics. In Section 3.2, we conduct a systematic temporal attention analysis on real videos, which leads us to

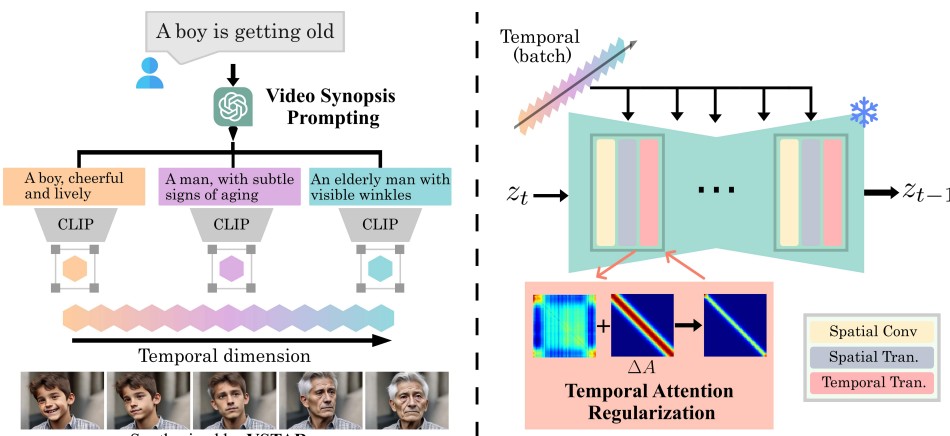

Figure 2: **Method overview.** VSTAR consists of two strategies: Video Synopsis Prompting (left) and Temporal Attention Regularization (right), where TAR is the key enabler for long video synthesis. We used color coding to indicate textual embedding interpolation for individual frames.

a simple yet effective Temporal Attention Regularization (Section 3.3), encouraging the temporal attention of synthetic videos to mimic the attention of real videos. We describe Video Synopsis Prompting in Section 3.4, which expands the initial text prompt for the whole video into several detailed descriptions that control the video progression respectively on different frames.

### 3.1 PRELIMINARY: TEXT-TO-VIDEO DIFFUSION MODEL

Most open-sourced text-to-video (T2V) diffusion models (Wang et al., 2023b;c; Chen et al., 2023; 2024a) share a similar high-level design, even if training strategies and specific implementations vary. Based on the text-to-image (T2I) latent diffusion model, e.g., Stable Diffusion (SD) (Rombach et al., 2022), two main changes are introduced for video diffusion models: inflating the 2D UNet to a 3D UNet and adding temporal transformers to capture the requisite temporal relationship found between video frames. With the addition of a temporal axis to the 2D convolutional kernels of SD, the resulting pseudo-3D convolutional layers can handle the input video latent $z \in \mathbb{R}^{N \times C \times H \times W}$, where $N$ is the number of frames and $C, H, W$ represent the channel and spatial dimension of each frame in the latent space, respectively. To generate a video of $N$ frames given a text prompt, these T2V methods (Wang et al., 2023b;c; Chen et al., 2023; 2024a; Guo et al., 2024) process all $N$ frames within one batch, and simply repeat the same prompt embedding for all frames. Inherently, the provided text prompt is conditioned via cross-attention of the spatial transformer in the UNet. The temporal transformer consists of several self-attention layers that operate along the temporal axis. More specifically, the spatial dimension of the intermediate features is merged into the batch dimension, resulting in a shape of $(B \times h \times w, N)$. Since the spatial layers inherited from SD can only handle each frame independently, the temporal attention layers thus play a crucial role for modeling the video dynamics.

### 3.2 TEMPORAL ATTENTION ANALYSIS

To properly synthesize videos that capture the dynamics conveyed in the input prompt, we delve into the synthesis model itself. An examination of the components new to T2V models, beyond the common building blocks already used in T2I models, leads naturally to the temporal attention layers. These new modules are crucial for facilitating proper video synthesis, i.e., generating sequential frames with dynamic yet consistent content that reflect the input text information. We hypothesize that the ineffectiveness of current T2V models arises from unstructured interactions among frames in the same video within the temporal attention layers. To verify our hypothesis, we conduct a systematic analysis comparing the attention maps of real and synthetic videos. Specifically, the attention map A is expressed as:

$$A = Softmax\left(\phi\left(Q, K\right)\right) = Softmax\left(\frac{QK^T}{\sqrt{d}}\right) \in \mathbb{R}^{N \times N},\tag{1}$$

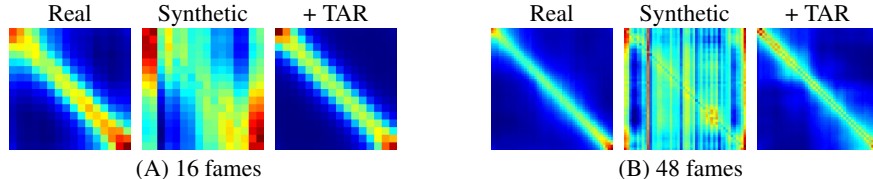

Figure 3: Temporal attention visualization of real and synthetic videos of 16 and 48 frames using VideoCrafter2. Attention of real videos exhibits a band-matrix like structure, indicating high correlation with adjacent frames. Synthetic videos exhibit less structured attention maps, especially for 48 frames, which explains the low quality of long video generation. The proposed Temporal Attention Regularization (TAR) can make the attention more structured, resembling the attention of real videos.

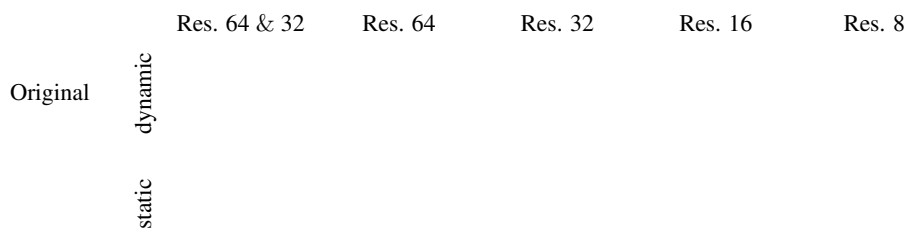

Figure 4: Per-layer temporal attention analysis. We replace the temporal attention maps at different resolutions with a diagonal matrix (1st row) and an all-ones matrix (2nd row), which leads to a more dynamic or a more static video, respectively. We observe that high resolution attention has a larger impact on the video dynamics. Note that this is a GIF, best viewed in *PDF Reader*.

where $Q$ and $K$ represent the query and key of the self-attention layer, and $d$ is the latent dimension. This attention matrix essentially depicts the pairwise correlation between the $N$ frames of one video. For real videos, their attention maps can be obtained by adding noise to their clean latent and extracting the attention during the denoising process. For synthetic videos, we can read out their attention maps directly during their synthesis passes.

For our analysis, we use VideoCrafter2 (Chen et al., 2024a) along with videos from the DAVIS dataset (Perazzi et al., 2016) and additional videos collected from the web. As shown in Fig. 3, for both 16 and 48 frame real videos, the attention matrix forms a band-matrix-like structure, with closer frames showing higher correlation to maintain temporal coherence. Compared to real videos, attention matrix of the synthetic ones is less structured, especially for 48 frames. That explains why the model generalizes even worse to longer videos. High correlation is spread across a wide range of frames, resulting in a harmonized sequence with similar appearances.

To understand the effect of temporal attention layers at different resolutions, we further conducted a per-resolution ablation as shown in Fig. 4. We replace the attention map at each individual resolution, i.e., 64, 32, 16, and 8, while keeping the other resolution untouched. We experiment with two extreme cases: using the Identity matrix ($I_N$) and the all-ones matrix ($J_N$). The former regularizes the frames to be mutually independent, while the latter oppositely requires full correlation, i.e., static sequence. The observations from Fig. 4 are highly consistent. When utilizing $I_N$ to encourage independence among frames, the temporal coherence of the synthesized frames is indeed compromised. Conversely, employing $J_N$ can significantly diminish the video dynamics, leading to a quasi-static video. This controlled experiment clearly demonstrates how the temporal attention layer impacts the dynamics of the video synthesis model.

Finally, we investigate the effect of the interplay between attention and resolution on the content dynamics of videos. As also shown in Fig. 4, the replacement at the higher resolutions of 64 and 32 has a more evident effect than at lower resolutions. Applying the changes jointly at both resolutions, 64 & 32, further amplifies the effect. In contrast, the videos are much less responsive to the attention replacement at resolution 8. Likely, the low resolution features encode high-level semantics, while with higher resolution features there is more capacity for representing varying local details in the scene; such details are necessary for reflecting coherent change over frames.

"A polar bear playing drum kit in NYC Times Square"

VideoCrafter2  **+ TAR**  VideoCrafter2  **+ TAR**  VideoCrafter2  **+ TAR**

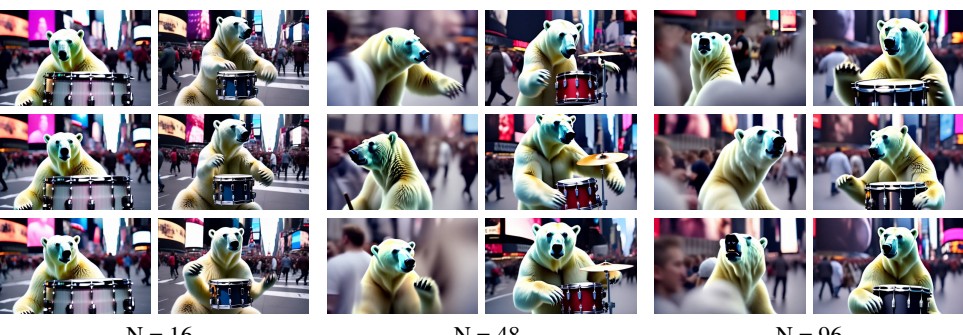

N = 16      N = 48      N = 96

Figure 5: TAR enables faithful longer video generation without training. VideoCrafter2 exhibits degradation in generating longer sequences, beyond 16 frames. Employing TAR leads to appealing results with improved dynamics. All results are generated with a *single* prompt. 1st row is a GIF.

Based on these controlled experiments, we can conclude that manipulating temporal attention allows us to alter the video dynamics, i.e., making the visual process either more static or more dynamic. In particular, adjustments at higher resolutions, e.g. 64 & 32, are more effective.

## 3.3 TEMPORAL ATTENTION REGULARIZATION (TAR)

The experiments above highlight the importance of temporal attention layers in determining video dynamics. Naturally, the attention matrices of synthetic videos should be similar to that of real videos. Therefore, we propose a simple regularization technique applied on the temporal attention layers for pretrained T2V model. Note that, our proposal is directly applied to pretrained T2V models without requiring re-training, and incurs no additional optimization costs during inference.

As illustrated in Fig. 3, the attention correlation of the real video resembles a band matrix, with high correlation between neighboring frames and lower correlation the larger the frame offset. To approximate such a structure, we design a symmetric Toeplitz matrix as the regularization matrix $\Delta A$, with its values along the off-diagonal direction following the Gaussian distribution:

$$\Delta A_{i,j} = e^{-\frac{1}{2}\left(\frac{j-i}{\sigma}\right)^2}, \tag{2}$$

where $i, j \in \{1, ..., N\}$ represent the entry index of the attention regularization map, and $\sigma$ is the standard deviation of the normal distribution. As indicated in Fig. 9, the standard deviation $\sigma$ can control the regularization strength, i.e. larger $\sigma$ leading to less visual variations along the temporal dimension. This regularization matrix is then added to the original attention matrix in (1), i.e.

$$A' \leftarrow Softmax\left(\phi(Q, K) + \max[\phi(Q, K)] \cdot \Delta A\right). \tag{3}$$

To balance both terms, we additionally introduce $\max[\phi(Q, K)]$, which weights $\Delta A$ based on maximum in the attention matrix $\phi(Q, K)$. As illustrated in Fig. 2, the regularized attention map $A'$ will be inserted back for further processing. After employing TAR, the regularized attention becomes more structured, resembling the attention of real videos as presented in Fig. 3.

TAR effectively supports temporal nursing for longer video generation using pretrained T2V models using a single text prompt, as demonstrated in Fig. 5, while also introducing no optimization overhead. VideoCrafter2 (Chen et al., 2024a) by default can generate 16-frame clips, with minor variations, and visual degradation occurs for longer videos. Using the same prompt, TAR can produce longer videos with improved dynamics, closely adhering to the prompt — all without the need for additional training. We find temporal attention analysis to be a powerful tool for understanding the temporal modeling of video diffusion models and leverage it to analyze other T2V models in Section 5. We establish valuable connections to their architecture designs, and provide guidance for the future training of T2V models for long video generation.

## 3.4 VIDEO SYNOPSIS PROMPTING (VSP)

When desired video involves gradual visual changes (see 3rd row Fig. 1 and Fig. 7), a single text is often not descriptive enough. Recent T2I works (Betker et al., 2023; Yang et al., 2024a; Wang et al., 2024a) leverage the in-context learning ability of LLMs for region-wise prompting and compositional generation. We similarly explore LLMs for Video Synopsis Prompting (VSP), to provide key visual states to decompose the transition along the temporal axis, as shown in Fig. 6. More implementation details can be found in Appendix C. It is sufficient to generate text descriptions for the main event changes in a video, rather than for each frame. A text encoder e.g., CLIP text decoder (Radford

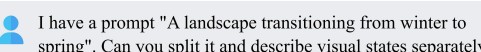
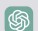

Figure 6: An illustration example of VSP. With LLMs, we can obtain more detailed descriptions for key stages.

et al., 2021), is then applied to extract the text embeddings of these descriptions, which are then interpolated to guide each frame's synthesis via cross attention as illustrated in Fig. 2. This process yields more accurate guidance for transitioning visual stages while ensuring smooth conditioning without abrupt changes between frames. VSP is a simple solution that can introduce more variation along the temporal axis, yet it alone cannot tackle the challenge of longer dynamic video synthesis, as seen in Fig. 7. Combined with TAR, which rectifies the temporal attention, our VSTAR is capable of synthesizing long sequences with appealing visual evolutions.

## 4 EXPERIMENTS

**Experimental setting.** To demonstrate the effectiveness of VSTAR in creating more dynamic videos, we run experiments and ablations on ChronoMagic-Bench-150 (Yuan et al., 2024) and prompts generated by ChatGPT (OpenAI, 2022) describing various visual transitions. These prompts are provided in the Supp. Material. We use CLIP-VL score for measuring alignment with the text prompt. Additionally, we employed the recent Metamorphic Score (MTScore) (Yuan et al., 2024) to measure the metamorphic amplitude of videos, where a higher MTScore indicates better video dynamics. More details can be found in Appendix A.3. By default, we employ the state-of-the-art open-sourced T2V model VideoCrafter2 (Chen et al., 2024a) with 320×512 resolution as our base model, which is combined with the proposed video synopsis prompting (VSP) and temporal attention regularization (TAR). We refer to this combination as our method or VSTAR throughout the experiments.

## 4.1 MAIN RESULTS

**Comparison with other T2V methods.** In Table 1, we provided a quantitative comparison between our VSTAR and SoTA T2V methods including ModelScope (Wang et al., 2023b), LaVie (Wang et al., 2023c), VideoCrafter2 (Chen et al., 2024a), Open-Sora-Plan V1.2 (Lab & etc., 2024) and CogVideoX-5B (Yang et al., 2024b), which requires a single inference pass, alongside FreeBloom (Huang et al., 2023) and FreeNoise[1] (Qiu et al., 2024), which utilize multiple passes, i.e., fusion of multiple inference runs. For a fair comparison, we adapted the first three models to support multi-prompt generation via VSP. However, this is not feasible with Open-Sora-Plan and CogVideoX, as they do not explicitly model the temporal dimension. For multi-pass methods, different prompts from VSP are naturally used in different inference runs. It can be seen that, our VSTAR achieves the best alignment with the text prompt, and more importantly, outperforms the others on MTScore by a large margin, demonstrating our advantage in generating dynamic visual evolution. Notably, enabled by TAR, our VSTAR can generate long videos in a single inference pass, without facing the challenges of maintaining the coherency of different passes.

We provide qualitative comparisons in Figs. 7, S.7 and S.8 for visual evolution synthesis. Although all methods generate meaningful results for 16-frame videos (see Figs. S.7 and S.8), the videos created by the other T2V models cannot properly reflect the visual content specified by the input prompts, i.e., lacking dynamic progression throughout the video. When generating 32 frames in one pass, as shown in Fig. 7, our method exhibits even greater advantages. The comparison methods yet again fail to produce the desired content, but this time to the extent that the visual quality of the individual frames

---

[1]We used the official implementation of FreeNoise which only supports two subprompts.

Table 1: Quantitative comparison with SoTA T2V methods. For a fair comparison, we adapted other methods to use Video Synopsis Prompting (VSP) when compatible. VSTAR outperforms the others, showing its strength in generating dynamic visual evolution, notably in a single inference pass.

| | ModelScope | LaVie | V.Crafter2 | Open-Sora-Plan | CogVideoX | FreeBloom | FreeNoise | **VSTAR** |
|---|---|---|---|---|---|---|---|---|
| VSP | ✓ | ✓ | ✓ | ✗ | ✗ | ✓ | ✓ | ✓ |
| Single-Pass | ✓ | ✓ | ✓ | ✓ | ✓ | ✗ | ✗ | ✓ |
| CLIP-VL↑ | 0.221 | 0.221 | 0.220 | 0.220 | 0.221 | 0.219 | 0.224 | **0.233** |
| MT-Score↑ | 0.322 | 0.306 | 0.401 | 0.239 | 0.312 | 0.361 | 0.379 | **0.448** |

"A landscape transitioning from winter to spring"

Figure 7: Comparison with other T2V methods on 32 frames generation, which is double the length of the default option. Our VSTAR can generate long videos with desired dynamics, while the others struggle to synthesize faithful results with visual progression using the same multi-prompt of VSP.

is also greatly compromised. In contrast, our VSTAR is able to generate longer videos with dynamic visual evolution, which is aligned with the quantitative evaluation in Table 1. More visual examples of VSTAR are presented in Appendix A.4, and qualitative results of Open-Sora-Plan and CogVideoX are provided in the Supp. Material. We also conducted a user study, detailed in Appendix A.2, where VSTAR clearly emerged as the preferred option.

**Transferring dynamics from real videos via TAR** Instead of using the predefined matrix formulated in Eq. (2), we explore transferring dynamics from a reference video, by employing a DreamSim (Fu et al., 2023) based similarity matrix as $\Delta A$ for TAR. DreamSim is a recently proposed visual similarity metric that has been demonstrated to align closely with human judgment. In the 1st row of Fig. 8, the inter-frame DreamSim similarity of the real video closely mirrors the temporal attention maps of the real videos, indicating that it is well-suited for TAR. In the 2nd and 3rd rows, we compare the results before and after applying DreamSim of the real video provided in the 1st row for TAR, using the same text prompt and random seed. As shown, after employing TAR with the DreamSim matrix, the video dynamics are noticeably enhanced, resembling the reference video, i.e., the gradual appearance of the rainbow. Meanwhile, the inter-frame similarity matrix of the synthesized video, shown on the right, aligns more closely with that of the real video. This demonstrates that dynamics from a reference video can be effectively transferred in a user-friendly manner to improve the generated output. Additionally, we use DreamSim for quantitative comparison of T2V models in Appendix A.1.

## 4.2 ABLATION STUDY

**Ablation on the effect of TAR and VSP.** We investigate the effects of the proposed Temporal Attention Regularization and Video Synopsis Prompting quantitatively in Table 2 using ChronoMagic-Bench-150. Employing VSP alone improves the CLIP-VL score slightly,

"Rainbow starts to appear after the rainy day"    DreamSim

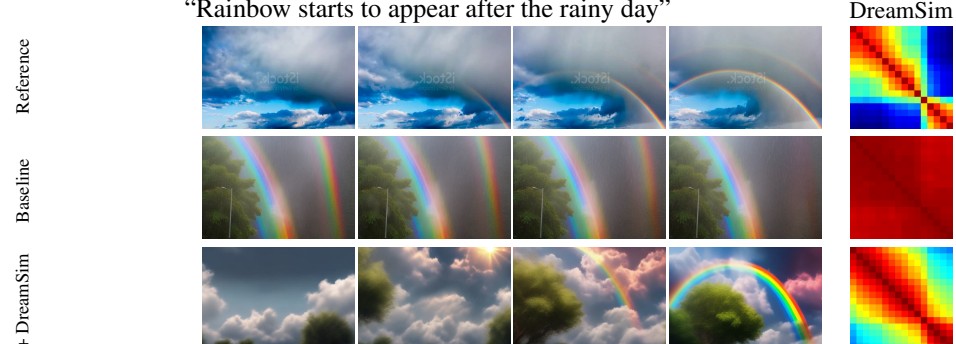

Figure 8: Transferring dynamics from real reference video (1st row) via employing its inter-frame DreamSim matrix in TAR enhances the output's dynamics as desired (3rd row).

"A peony starts to bloom, in the field"

| None | $\sigma_{64} = 8$ | $\sigma_{64} = 4$ | $\sigma_{64} = 1$ | $\sigma_{64} = 1$ $\sigma_{32} = 8$ | $\sigma_{64} = 1$ $\sigma_{32} = 1$ |

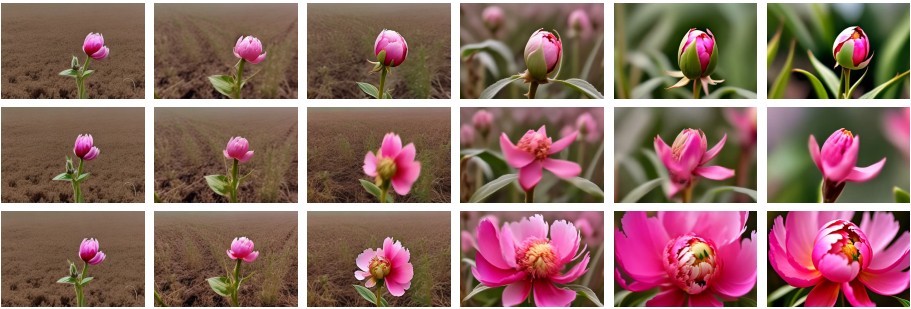

Figure 9: Ablation of attention regularization matrix $\Delta A$. Smaller $\sigma$ induces a stronger regularization effect, leading to increasing temporal dynamics. When applying regularization at both 64 & 32, the video becomes more dynamic, i.e., the peony is fully bloomed. 1st row is a GIF.

but brings little help on introducing more visual changes, i.e., minor difference on MTScore. This is aligned with our qualitative results in Appendix A.5. Despite VSP providing a more descriptive summary of different visual states, without TAR, the temporal attention remains strongly correlated, yielding limited visual dynamics. When combining both strategies, our VSTAR can effectively synthesize the desired visual content, exhibiting improved dynamics with a more appealing time-lapse effect. For prompts that do not require gradual visual appearance changes, VSP is not necessary. However, TAR still plays a crucial role in enabling high quality longer video generation, as shown in Figs. 1 and 5.

Table 2: Ablation on VSP and TAR.

| | CLIP-VL↑ | MT-Score↑ |
|---|---|---|
| Baseline | 0.214 | 0.397 |
| + VSP | 0.220 | 0.401 |
| + VSP & TAR | **0.233** | **0.448** |

**Ablation on regularization matrix.** We further ablate by investigating the effect of using a different standard deviation $\sigma$ in the regularization matrix $\Delta A$, shown in Fig. 9. We start from applying regularization at the highest temporal resolution i.e., 64, since high-resolution temporal attention more greatly influences the video dynamics, as demonstrated in the temporal attention analysis in Section 3.2. The results show that decreasing $\sigma$ results in a stronger regularization effect, inducing more pronounced visual changes throughout the video (e.g. compare row 2 to row 4, and notice the extent of the blooming of the flower). Going one step further, applying regularization also at a resolution of 32 results in the peony reaching its fullest bloom. However, when equally strong regularization is applied at both a resolution of 64 and 32, i.e., $\sigma_{64} = \sigma_{32} = 1$, the visual changes can be too excessive, leaving the impression of poor temporal coherency across frames. Empirically, we find that applying $\sigma_{64} = 1$ strikes a good balance between dynamic changes and temporal coherency, which is our default option.

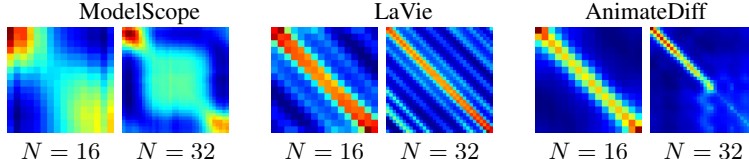

Figure 10: Temporal attention visualization of other T2V Models for the default 16 frame and longer 32 frame videos. ModelScope faces similar issues as VideoCrafter2 (see Fig. 3), with high correlation spread across many frames, especially for $N = 32$. LaVie and AnimateDiff, using frame index positional encoding, struggle to generalize beyond the 16-frame limit they were trained on.

## 5    DISCUSSION

VSTAR offers a simple yet effective training-free solution for longer dynamic video synthesis. Yet, certain limitations of the pretrained T2V models, e.g., poor understanding of long-range motion, can be further addressed during training. Therefore, we next analyze the temporal attention units of various T2V models to offer insights for improving architecture design and training strategies.

In Fig. 10, we visualize the temporal attention layers of ModelScope (Wang et al., 2023b), LaVie (Wang et al., 2023c) and AnimateDiff (Guo et al., 2024). It can be seen that ModelScope exhibits similar attention behavior to VideoCrafter2 (see Fig. 3), with temporal correlation deteriorating in longer videos, even at 32 frames, i.e., twice the standard length. This aligns with the qualitative comparison in Fig. 7. In Appendix A.6, we show that VSTAR can also improve ModelScope's long video generation. AnimateDiff and LaVie demonstrate different temporal attention behavior, due to the incorporation of Rotary Positional Encoding (Touvron et al., 2023) in the former and Sinusoidal Positional Encoding in the latter. With the positional encoding, the models learn better temporal correlation among neighboring frames for 16 frames, showing a band-matrix structure more closely resembles that of real videos. However, when generating videos longer than its training capacity, the model faces considerable difficulty in preserving the desired temporal dynamics, resulting in inferior synthesis quality, as depicted in Fig. 7. The Rotary Positional Encoding employed in LaVie is a form of relative positional encoding, i.e., it depends on the relative offsets of frames, which could explain the periodic pattern seen in the attention maps. While the Sinusoidal Positional Encoding used in AnimateDiff is based on the absolute frame index, leading to the model failing completely for indices unseen during training (past 16). Transformer-based models often rely on positional encodings, which largely constrain their generalization ability for longer video synthesis. These observations concerning T2V models are interestingly aligned with prior studies regarding Positional Encoding on length generalization in Transformers (Kazemnejad et al., 2023) in the context of LLMs.

This comparison offers valuable insights into improving the training of the next generation of T2V models. For instance, omitting positional encoding can improve generalization capability, and incorporating a regularization loss on the temporal attention maps can help to enforce the desired temporal dynamics. Alternatively, one can employ a better combination of data format and positional encodings, as explored in the recent work (Zhou et al., 2024), which achieves improved length generalization. For instance, Randomized Positional Encoding (Ruoss et al., 2023) can help to avoid overfitting on the position indices, and mixing up subsampled video sequences can further strengthen local correlations. Combining such techniques with our VSP and TAR may further improve long video generation with better long-range understanding.

## 6    CONCLUSION

In this paper, we propose a training-free Generative Temporal Nursing method - VSTAR that enables the generation of longer (e.g. $32 \sim 128$ frames) and temporally coherent appealing videos in a single inference pass. As our core contribution, Temporal Attention Regularization can effectively improve the temporal modeling of pretrained T2V models, especially for longer video synthesis. Additionally, TAR offers flexible control over video dynamics. Besides motivating TAR, our analysis of temporal correlation in real videos offers valuable insights into improving the design and training of the next generation of T2V models. For example, some form of positional encoding appears to be hampering generalization capability, while the incorporation of a regularization loss on temporal attention maps can help to enforce temporal dynamics. While VSTAR is readily applied to pretrained T2V models, future work may incorporate it during training for improved procedural dynamics, such as complex activities on respective data.

ETHICS STATEMENT

We have carefully read the ICLR 2025 code of ethics and confirm that we adhere to it. The proposed method is training-free and built upon pretrained T2V models, thereby no further privacy concerns should arise. Nevertheless, given the imbalanced nature of large-scale datasets used to train the original T2V models, these pretrained models may inherit certain data biases, inaccurately representing the diversity of the overall population. These biases can potentially reinforce existing societal stereotypes and inequalities. Therefore, it is advisable to undertake proactive steps to identify and mitigate such biases, which may include the involvement of human reviewers in sensitive contexts.

REPRODUCIBILITY STATEMENT

Regarding reproducibility, our implementation and experiments are based on publicly available models Wang et al. (2023c;b); Guo et al. (2024); Chen et al. (2024a); Yang et al. (2024b); Lab & etc. (2024). Details on the experimental settings are given at the beginning of Section 4 and Appendix A. We plan to release the code upon acceptance.

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

# VSTAR: Generative Temporal Nursing for Longer Dynamic Video Synthesis
## Supplementary Material

This supplementary material to the main paper is structured as follows:

- In Appendix A, we provide more experimental results, including quantitative comparisons, a user study, additional visual results, ablation on the effect of TAR and VSP, and the combination of VSTAR and ModelScope. We also include more details on the MTScore evaluation.

- In Appendix B, we discuss our attempts and insights into optimization-based temporal generative nursing, which might spark interest for subsequent studies.

- In Appendix C, we elaborate further on Video Synopsis Prompting., e.g., how to instruct LLMs to generate the video synopsis.

- In Appendix D, we provide additional visualizations of the regularization matrix and samples of collected real dynamic videos.

- In Appendix E, we discuss limitations of the proposed method.

In https://github.com/YumengLi007/VSTAR_ICLR2025_Supp, we provide more visuals including:

- Qualitative results of SOTA T2V models, i.e., Open-Sora-Plan-V1.2 and CogVideoX-5B.
- GIF version of our VSTAR's qualitative results.
- Effect of TAR on longer video generation.
- Qualitative comparison with multi-pass methods.
- Additional ablation results of attention regularization matrix.
- Samples of real videos.

## A    MORE EXPERIMENTAL RESULTS

### A.1    QUANTITATIVE COMPARISON

We quantitatively compute the similarity of two frames a certain interval apart using the recent perceptual similarity metric DreamSim (Fu et al., 2023). In Fig. S.1, we plot the similarity matrices of real videos from DAVIS (Perazzi et al., 2016) and web sources, alongside those synthesized by VideoCrafter2 and our VSTAR. The values in the matrix are normalized across all methods. VideoCrafter2 has an overall high similarity across all frames, which is aligned with the visual outcomes containing limited visual variations. In contrast, our VSTAR mirrors the perceptual similarity patterns of real videos, demonstrating the effectiveness of our approach in enhancing video dynamics. The similarity distribution of is presented in Fig. S.2. For desired video dynamics, the similarity should decrease as the interval between frames increases, signaling a steady visual evolution. Meanwhile, frames that are closer together should exhibit higher similarity compared to those further apart, indicating preserved temporal coherence. The distribution exhibited by VideoCrafter2 is highly concentrated at the high similarity region, even with large intervals. This can be explained by the fact that it often generates videos with limited visual variation over time, which is aligned with the qualitative results and analysis in Sec. 4.1 of main paper. In contrast, with an increasing interval between frames, the distribution for both our VSTAR and real dynamic videos slightly shifts towards a region of lower similarity, indicating that more visual variation has been introduced within the video. Our distribution extensively overlaps with that of real videos, suggesting that our results not only exhibit improved temporal dynamics, but also maintain the continuity.

### A.2    USER STUDY

For further evaluation, we conducted a user study to compare our VSTAR with the SOTA T2V model VideoCrafter2 (Chen et al., 2024a). 110 individuals with diverse backgrounds participated in the user

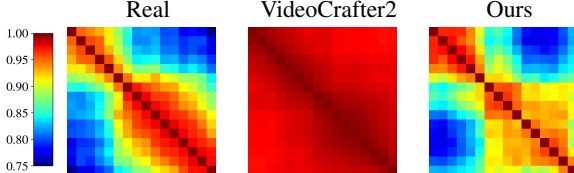

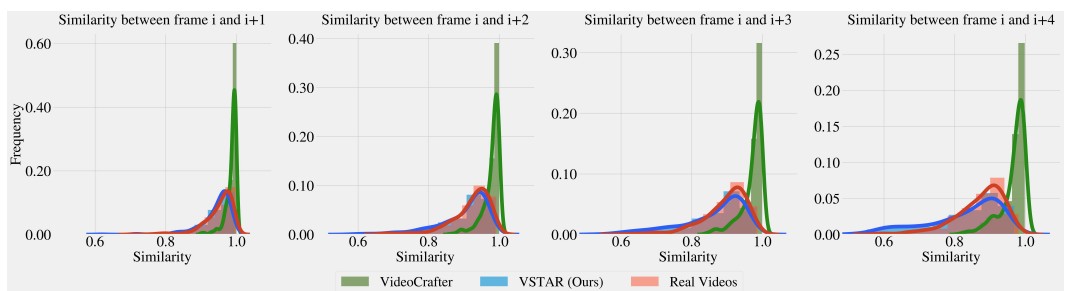

Figure S.1: Inter-frame perceptual similarity matrix based on DreamSim (Fu et al., 2023), where values are normalized across *all* methods. VideoCrafter2 has high similarity across nearly all frames, which is aligned with the visual results lacking variation. In contrast, our synthesized videos highly resemble the real ones, indicating desired dynamics.

Figure S.2: Comparison of DreamSim Similarity Distribution between real videos, VideoCrafter and our VSTAR. For preferred video dynamics, the similarity should decrease as the interval between frames increases, signaling a steady visual evolution. Meanwhile frames that are closer together should exhibit higher similarity compared to those further apart, indicating sustained temporal coherence. For VideoCrafter2, the majority is located in the high similarity region regardless of the interval, indicating that there is limited visual variation within the video. The distribution of ours overlaps significantly with that of real videos, suggesting improved video dynamics without compromising coherency.

study, working in fields such as computer vision, reinforcement learning, natural language processing, art design, medical engineering, mechanical engineering, and administrative management, among others. We assess the videos across four dimensions: text alignment, video dynamics, visual quality and temporal coherency. Text alignment concerns whether the synthesized results properly reflect the input text prompt. Video dynamics examines the dynamic visual changes within the progression of the video. A higher visual quality indicates fewer artifacts and distortions, leading to a more visually pleasing result. Temporal coherency evaluates if the result is temporally smooth, i.e., there are no abrupt or unexplained changes that could disrupt the viewing experience. For the first three aspects, participants are presented with paired results to evaluate, selecting one over the other or deeming them equivalent. Regarding temporal coherency, we pose a simple yes-or-no question, asking whether the participants perceive the video as being temporally smooth.

The outcome is summarized in Fig. S.3. Our VSTAR emerges as the preferred choice across various frame lengths from all aspects, with its advantages becoming more pronounced in the generation of longer videos with $N = 32 \sim 64$. Importantly, our method not only enhances video dynamics but also preserves temporal coherency. A majority of participants confirmed that our results exhibit smooth temporal transitions, with $87.6\%$ for standard-length videos and $79.1\%$ for longer videos agreeing to this assessment. This favorable reception surpasses the baseline VideoCrafter2, possibly as a result of its less engaging content.

Additionally, we included pairs of videos, both generated by VSTAR, to verify the consistency of our method's improvement, making it challenging for users to make a clear choice. As shown in Fig. S.4, participants indeed often found it difficult to differentiate, with $52.7\%$, $40.3\%$ and $50.9\%$ of them rating both videos as equal in terms of text alignment, visual dynamics, and visual quality, respectively. The remaining participants were divided in their preference between the two videos. This indicates that our synthesis results are consistent and display a narrow gap between them.

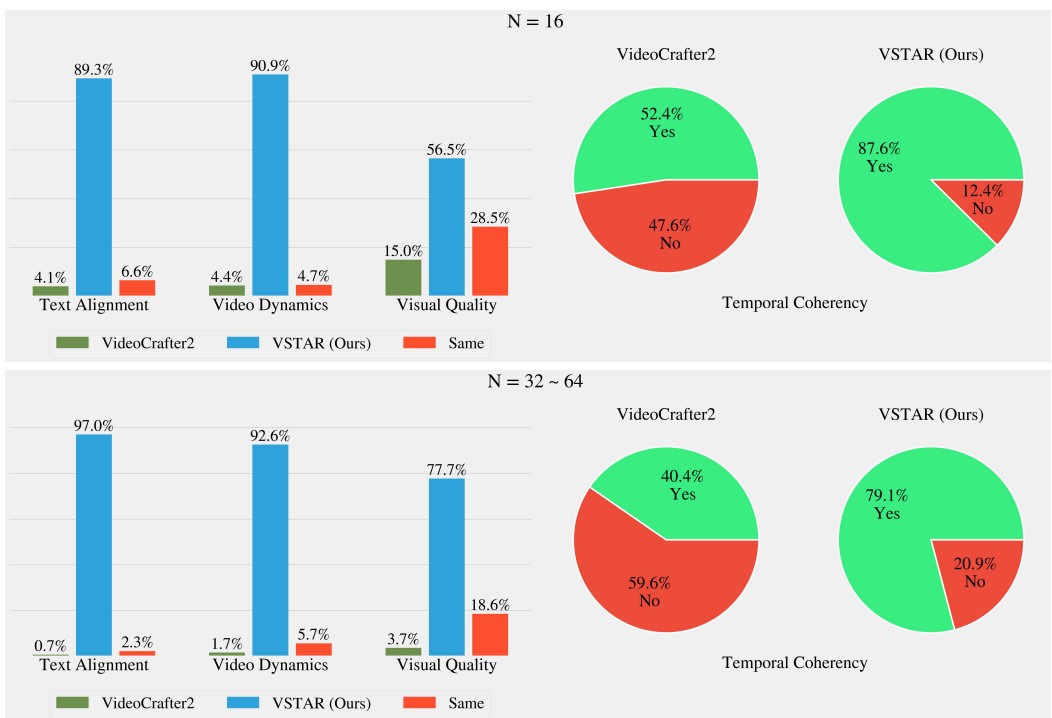

Figure S.3: User study on both standard 16 frames and longer videos with $32 \sim 64$ frames. For the first three aspects, participants review pairs of videos, choosing between them or rating them as the same. For temporal coherency, the numbers are the absolute probability that a participant perceives the video from the respective method as having smooth temporal progression.



Figure S.4: User study on paired of videos, both generated by our VSTAR, to verify the consistency of our method's improvement, making it challenging for users to make a clear choice. Indeed, a large number of participants perceived both videos as identical across all three aspects. The rest had diverse preferences between the two videos. This demonstrates the consistency of our synthesis results and their closely matched quality.

## A.3    MTSCORE EVALUATION

Metamorphic Score (MTScore) (Yuan et al., 2024) is a newly proposed evaluation metric designed to assess the metamorphic amplitude within generated videos, specifically measuring the extent and degree of visual transformations over time. Previous metrics such as CLIP related scores focus on the vision-language alignment and temporal coherence, however, they overlook the crucial aspect of visual evolution within generated videos. For instance, prior methods often tend to generate quasi-static content with minimal visual change across frames, which cannot be properly reflected in CLIP scores. MTScore, therefore, addresses this gap by directly measuring the degree of visual metamorphosis leveraging the video retrieval model.

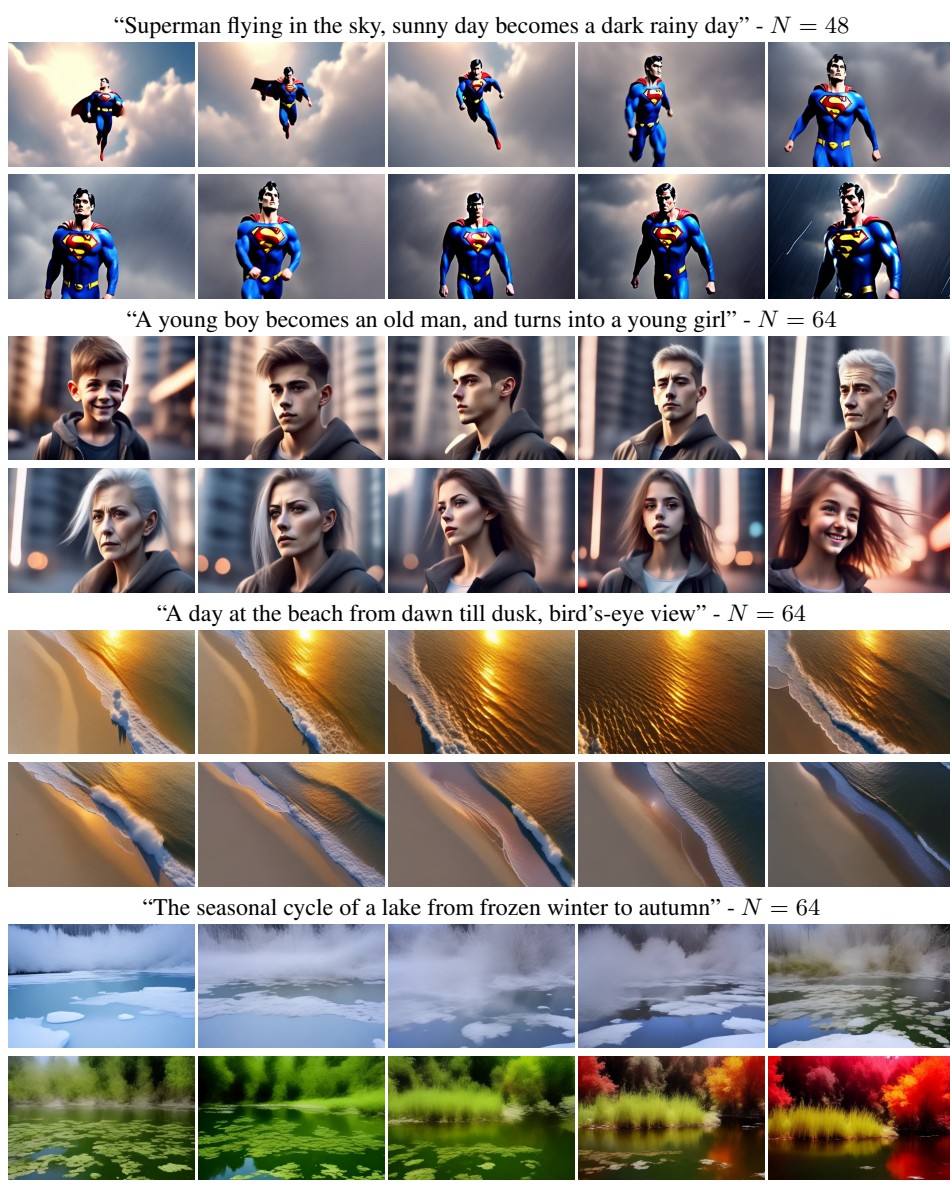

"Superman flying in the sky, sunny day becomes a dark rainy day" - $N = 48$

"A young boy becomes an old man, and turns into a young girl" - $N = 64$

"A day at the beach from dawn till dusk, bird's-eye view" - $N = 64$

"The seasonal cycle of a lake from frozen winter to autumn" - $N = 64$

Figure S.5: Qualitative results of videos with 48 and 64 frames synthesized by VSTAR. Images are sub-sampled from the sequence. GIF version is provided in G.2.

More specifically, Yuan et al. (2024) designed ten retrieval texts to differentiate between progressing and normal videos, e.g., "A normal video, not a time-lapse video" and "A time-condensed video, not a conventional video". Using multiple retrieval sentences enhances the evaluation robustness and accuracy. These text prompts, alongside the videos, are fed into a video retrieval model, i.e., InternVideo2 (Wang et al., 2024b), which yields the probabilities for being categorized into metamorphic or general videos. Therefore, a higher MTScore indicates the video demonstrates a more pronounced and dynamic transformation over time, effectively capturing visual evolution. As shown in Table 1, our VSTAR outperforms the other SOTA T2V models on MTScore by a large margin, highlighting its strength in synthesizing dynamic videos.

## A.4 MORE VISUAL RESULTS

More qualitative visual results are provided in Figs. S.5 and S.6, in which the length of videos varies from the default 16 frames to longer ones with 64 frames. Intriguingly, all videos are generated in

"A mural being painted on a city wall"

"The process of a woman losing 50kg of weight"

"A caterpillar transforming into a butterfly"

"A female person's hairstyle changing through the years"

"A pizza is being made"

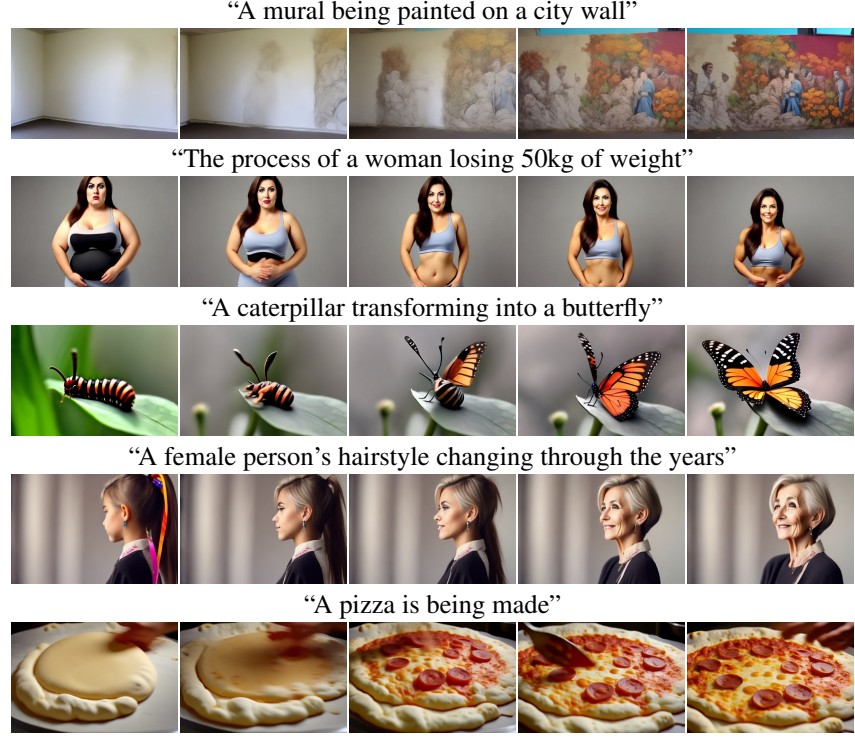

Figure S.6: Qualitative results of videos with 16 frames synthesized by VSTAR. Images are sub-sampled from the sequence. GIF version is provided in G.2.

"A Ferrari driving on the road, starts to snow"

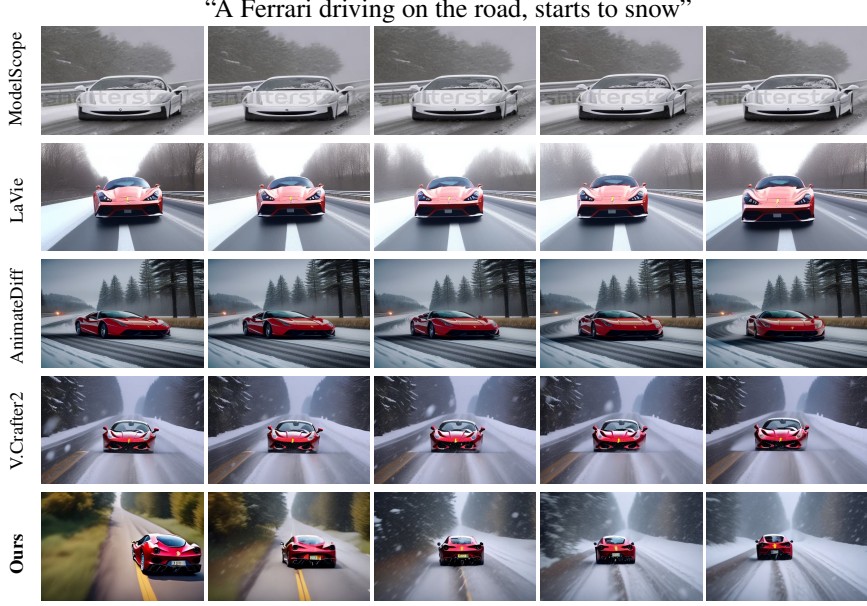

Figure S.7: Comparison with other T2V models on 16 frames generation. Our VSTAR can synthesize desired visual development from a clear day to snowy scene, while the others tend to generate the final visual state, i.e., snowy day. GIF version is provided in G.4.

one *single* pass using our VSTAR. Additional results on the comparison with other T2V methods can be found in Fig. S.8. It can be seen that our VSTAR consistently outperforms the other T2V models, demonstrating better dynamics with more visual changes over time complying with the text prompt.

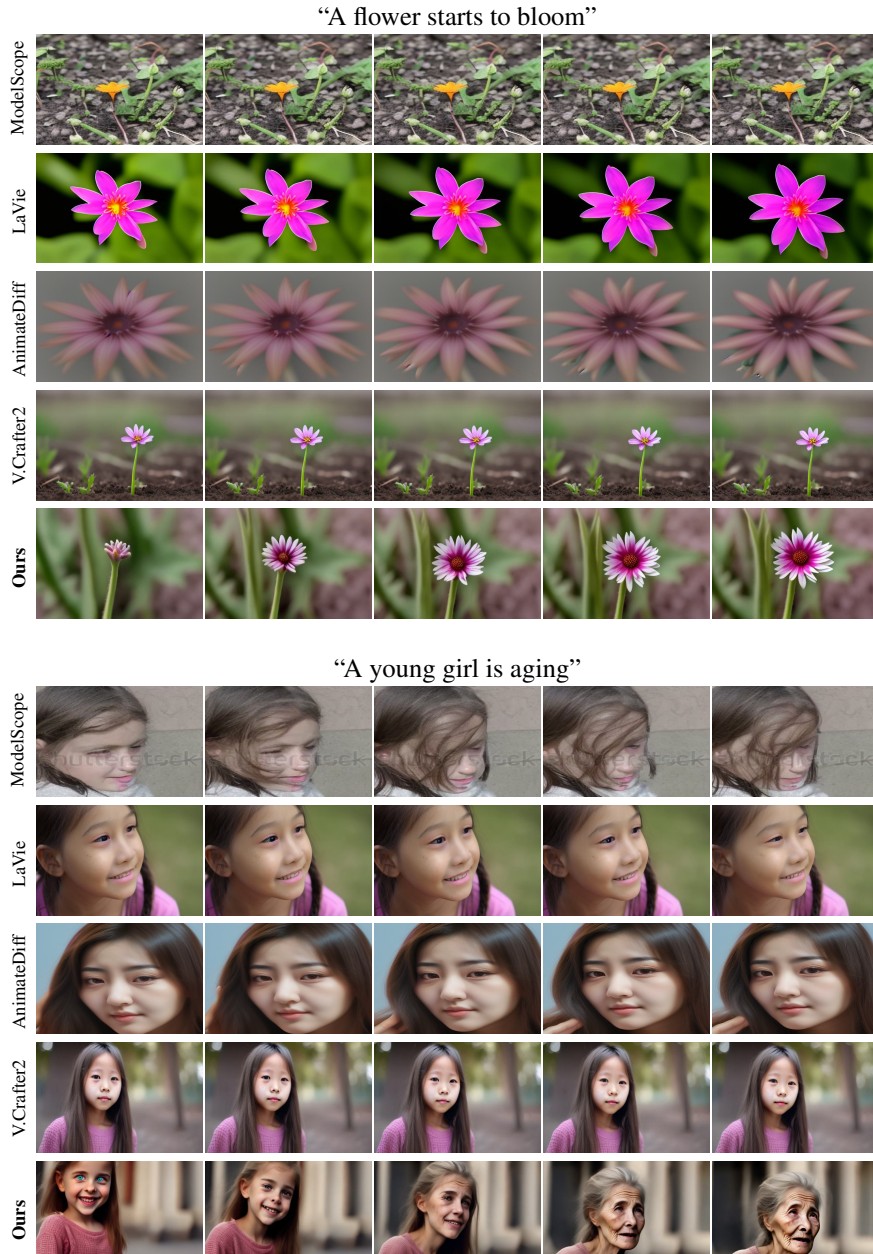

Figure S.8: Comparison with other T2V models on 16 frames generation. Our VSTAR consistently demonstrates improved video dynamics, resulting in more visually appealing content compared to the other methods. GIF version is provided in G.4.

## A.5 ABLATION ON THE EFFECT OF TAR AND VSP

We investigate the effects of the proposed Temporal Attention Regularization and Video Synopsis Prompting individually for dynamic visual evolution generation in Fig. S.9, where we generate videos of 48 frames in one pass based on the prompt "Spiderman on the beach from morning to evening", using the same initial noise. The synthesized video clips are presented in the first column as GIFs; the other images are subsampled from the full sequence. The baseline model VideoCrafter2 struggles to synthesize a video faithful to the input prompt, generating a sequence of highly similar frames, with a stride-like texture in the background, that fail to depict the time-lapse video. When employing the

"Spiderman on the beach from morning to evening"

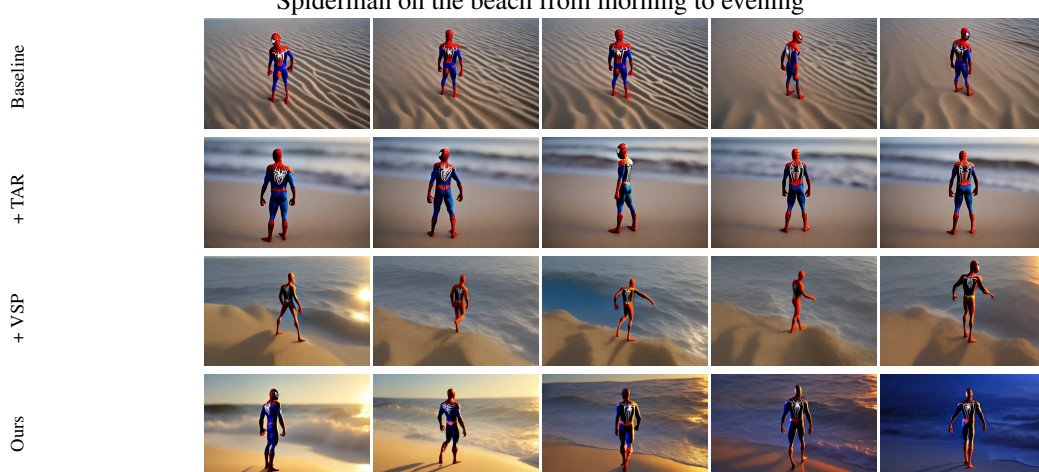

Figure S.9: Ablation on the effect of Video Synopsis Prompting (VSP) and Temporal Attention Regularization (TAR). Subsampled from 48 frames. Combination of TAR and VSP effectively enables long video generation with desired visual evolution. While individual strategy improves upon the baseline, there still lacks of desired dynamics.

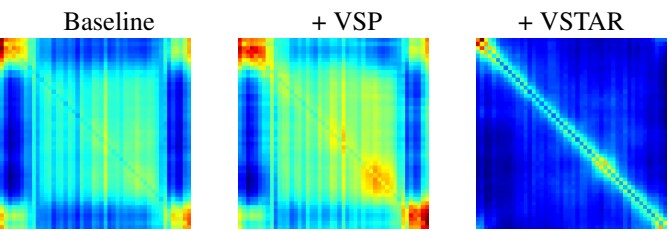

Figure S.10: Comparison of temporal attention maps between using and without using TAR. It can be seen that using VSP alone does little to improve the temporal attention structure while employing TAR is more effective.

TAR, the model generates a more realistic sequence, however without the desired visual evolution; the single plain prompt is insufficient to describe the scene changes. Interestingly, while VSP provides a more descriptive summary of different visual states, without TAR, the temporal attention remains strongly correlated. The model then attempts to depict the provided textual description, however with limited visual variation. When combining both strategies, our VSTAR can effectively synthesize the desired visual content, exhibiting improved dynamics with a more appealing time-lapse effect. In Fig. S.10, we compare the temporal attention maps when using different strategies. It can be seen that using VSP alone does little to improve the temporal attention structure. In contrast, when combined with TAR, the temporal attention becomes more structured, and the synthesized results align with the desired content. This highlights the crucial role of TAR in generating long dynamic videos.

For the input text prompts that do not require gradual visual appearance changes within the video, VSP is not necessary. However, TAR still plays a crucial role in enabling high quality longer video generation. In Fig. 5 and G.3, we provide visual comparison without and with TAR for longer video generation, employing a *single* text prompt without employing VSP.

## A.6 COMBINATION OF VSTAR AND MODELSCOPE

In the main paper, we by default apply proposed VSTAR with state-of-the-art open-sourced T2V model VideoCrafter2 (Chen et al., 2024a). Nonetheless, VSTAR can also be combined with other pretrained T2V models to enhance their video dynamics. In Fig. S.11, we showcase that VSTAR can boost the long video generation ability of pretrained ModelScope (Wang et al., 2023b), resulting in

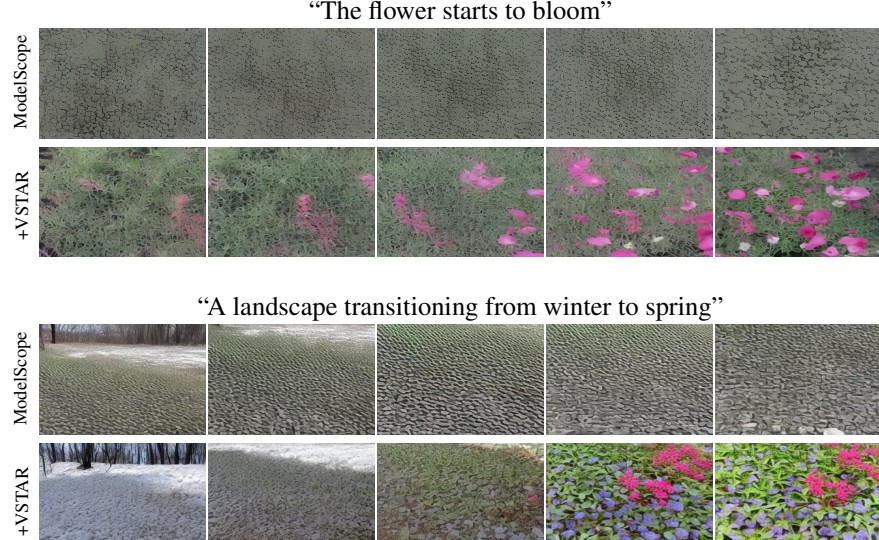

Figure S.11: Combination of ModelScope with VSTAR on 32 frames generation, which is double the length of the default option. The same random seed is used. ModelScope cannot generalize well to unseen frames, as discussed in Sec. 4.1. Applying our VSTAR can significantly boost its generalization ability without fine-tuning required.

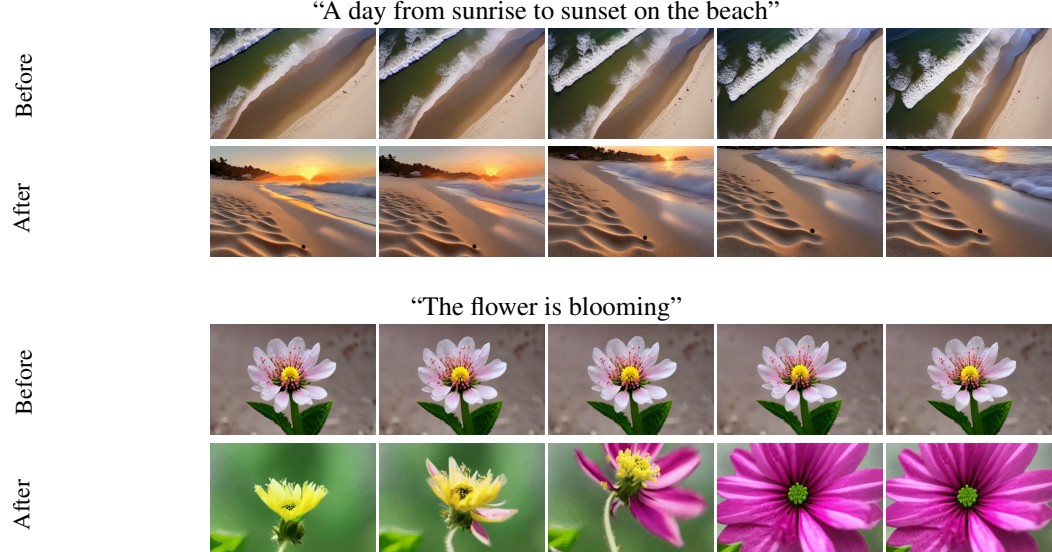

Figure S.12: Comparison of before and after applying initial noise optimization for 16 frames generation using the same single prompt and random seed. After optimization, the video dynamics has been enhanced, i.e., more visual variation has been introduced over time. Note that the first column is a GIF, best viewed in *Acrobat Reader*.

better visual quality and video dynamics. However, due to the constrained capacity of the base model ModelScope, the overall synthesis results underperform those achieved by combining VSTAR with VideoCrafter2 as shown in the main paper.

## B  OPTIMIZATION-BASED GENERATIVE TEMPORAL NURSING

As a method of generative temporal nursing (GTN), our VSTAR is completely training- and optimization-free, and can be readily applied to frozen pretrained T2V models without introducing inference time overhead. Additionally, we explored optimization-based GTN, assuming that a real reference video is available to guide the learning of desired dynamics. Inspired by the temporal attention analysis detailed in Sec. 3.3, we attempt to optimize the initial noise latents at inference time to align the attention maps of the given real video and the synthesized one, as outlined below.

Following Ma et al. (2024), we parameterize the initial video latents of $N$ frames with a Multivariate Gaussian distribution $\epsilon \sim N(\mu, \mathbf{\Sigma}_N(\beta, \gamma)))$, where $\mathbf{\Sigma}_N(\beta, \gamma)$ denotes the covariance matrix:

$$\mathbf{\Sigma}_N(\gamma) = \begin{pmatrix} \beta & \gamma & \gamma^2 & \cdots & \gamma^{N-1} \\ \gamma & \beta & \gamma & \cdots & \gamma^{N-2} \\ \gamma^2 & \gamma & \beta & \cdots & \gamma^{N-3} \\ \vdots & \vdots & \vdots & \ddots & \vdots \\ \gamma^{N-1} & \gamma^{N-2} & \gamma^{N-3} & \cdots & \beta \end{pmatrix}. \tag{5}$$

Given a real reference video, we can add noise to its clean latent and extract temporal attention maps $A_t^{ref}$ from its denoising process at the timestep $t$. Then, we perform an initial noise optimization to match the temporal attention maps $A_t$ during the synthesis process with that of the reference ones:

$$L_{Attn} = \left\| A_t^{ref} - A_t \right\|. \tag{6}$$

Furthermore, to prevent the initial noise from deviating significantly from the Gaussian Distribution, we minimize the Kullback-Leibler divergence between the optimized latents and the standard Gaussian Distribution:

$$L_{KL} = KL(N(\mu, \mathbf{\Sigma}||N(0, I)). \tag{7}$$

The joint optimization loss is a weighted sum of both loss terms:

$$\min_{\epsilon} L_{joint} = \min_{\mu, \beta, \gamma} L_{all} = \min_{\mu, \beta, \gamma} L_{attn} + \lambda L_{KL}, \tag{8}$$

where $\lambda$ is a weighting factor.

As shown in Fig. S.12, after applying the initial noise optimization, the temporal dynamics of synthesized results from the same single prompt have noticeably improved, with more visual changes occurring throughout the video's progression. However, this optimization-based technique increases the inference time and demands more memory, making it challenging to scale for longer videos. In this regard, VSTAR stands out as more scalable and efficient, demonstrating its capability for facilitating long video generation. Overall, we can see both approaches highlight that regularizing the temporal attention is an effective solution, suggesting that further exploration in this area could present an intriguing direction for future research.

## C  MORE DETAILS ON VIDEO SYNOPSIS PROMPTING

Leveraging the in-context learning capability (Brown et al., 2020; Hu et al., 2022) of LLMs, we can guide them to perform the video synopsis prompting task automatically through prompting with a single concrete example. For instance, we can instruct ChatGPT (OpenAI, 2022) with the following prompt:

> I have a prompt "A landscape transitioning from winter to spring" for video generation. Can you split the process and describe the states separately? Each state is described in only one sentence and please consider the coherency between sub-prompts. Please be straightforward and do not use a narrative style.
> For example, for prompt "a boy is getting old", it can be divided into two states, e.g., "a young boy" and "an old man".
> Based on this example, can you provide the description? The number of states is not limited to two.

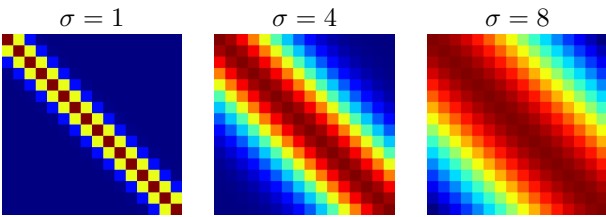

$\sigma = 1$      $\sigma = 4$      $\sigma = 8$

Figure S.13: Visualization of regularization matrix $\Delta A$ with different standard deviation $\sigma$. A Smaller $\sigma$ can enhance the effect of regularization.

Subsequently, ChatGPT can provide a detailed video synopsis that includes multiple visual states. Once the LLM has learned such a task, we can then simply prompt it to execute the task without reiterating the examples:

> I have a prompt "A peony starts to bloom, in the field". Can you split the process and describe the states separately?

Original prompts and the ChatGPT generated video synopsis are available in the *prompt_list.json* file included in the Supp. Material.

It is sufficient to generate text descriptions for the main event changes in a video rather than for each frame. A text encoder e.g., CLIP text decoder (Radford et al., 2021), is then applied to extract the text embeddings of these descriptions, which are then interpolated to guide each frame's synthesis via cross attention as illustrated in Fig. 1. This process yields more accurate guidance for transitioning visual stages, while ensuring smooth conditioning without abrupt changes between frames. All textual embeddings are fed to the model in a single inference pass.

## D  OTHER VISUALIZATION

### D.1  VISUALIZATION OF ATTENTION REGULARIZATION MATRIX

The regularization matrix $\Delta A$ is designed as a symmetric Toeplitz matrix with values along the off-diagonal direction following a Gaussian distribution. In Fig. S.13, we visualize $\Delta A$ with different standard deviations $\sigma$. We can see that as $\sigma$ decreases, the correlation increasingly concentrates on adjacent frames, thereby amplifying the regularization effect.

### D.2  EXAMPLES OF REAL VIDEOS

We provide some examples of real dynamic videos in G.7. They are collected from the web, DAVIS dataset (Perazzi et al., 2016), showcasing diverse content. The selected videos contain ample visual changes over time, as opposed to static clips, and they are all captured using a single-camera setup.

## E  LIMITATIONS

VSTAR offers a simple yet effective solution for improving pretrained T2V models, however, there are fundamental issues of pretrained models that may not be completely resolved via generative nursing at inference time only. Although our VSTAR has eased the process of reasoning prompts that involve dynamic evolution, the model can still struggle with responding to the decomposed open-world prompts, resulting in visuals that are not aligned with the prompt, potentially due to limited capability of the text encoder (Podell et al., 2024; Liu et al., 2024). Nevertheless, several recent works (Chefer et al., 2023; Li et al., 2023) have employed on-the-fly latent optimization to improve the textual alignment of a frozen T2I model. One may explore the combination of VSTAR with such techniques for further improvement. Due to the limited capability of pretrained T2V models in motion generation, the results may occasionally appear less natural. Nevertheless, we are

optimistic that the community will continue to develop and open-source better and larger models trained on more extensive datasets, further enhancing motion quality.

