# OpenReview forum: "VSTAR: Generative Temporal Nursing for Longer Dynamic Video Synthesis"
_ICLR.cc/2025/Conference — ICLR 2025 Poster_

### Official Review · Reviewer_JuSf · 2024-10-21

**Soundness:** 3
**Presentation:** 3
**Contribution:** 3
**Rating:** 6
**Confidence:** 5

**Summary:**

In this paper, the authors first analyze the behavior of temporal attention when generating longer videos. Then based on the observation, they propose VSTAR, a tuning-free method for longer video generation. The framework supports the transition of mult-prompt through the planning of pre-trained LLMs.

**Strengths:**

1. The analysis of temporal attention has some inspiration.
2. The design of TAR enables longer video generation in a one-pass behavior.
3. Benefiting from one-pass generation, the framework naturally supervises the coherence of results described by VSP, making it easier to achieve the transition of mult-prompt than the previous methods.
4. Although the quantitative results in this paper are still to be completed, the qualitative results are relatively sufficient, fully explaining the role of each module of VSTAR and other methods compared.

**Weaknesses:**

1. This paper does not evaluate some classic metrics, such as FVD. Alternatively, evaluating on a comprehensive benchmark like VBench will make the comparison more convincing.
2. Table 2 lacks a setting of Baseline+TAR. Purely Baseline+VSP is supposed to bring obvious content mutation while gaining better scores compared to Baseline, further illustrating that the two indicators in the paper are not enough to fully evaluate the quality of video generation.
3. When comparing VSTAR with FreeNoise in a mult-prompt setting, noticeable flickering in FreeNoise exhibits the implementation is not aligned with the original paper. (Although this does not affect my recognition of the effectiveness of VSTAR in the mult-prompt setting.)

**Questions:**

Most concerns are listed in weakness. Out of curiosity, FreeLong also mentions a similar phenomenon in temporal attention. Can you briefly explain the pros and cons of the two designs? Of course, it is not necessary to discuss the concurrent work according to the Reviewer Guide.

---

> ### Author Response · Authors · 2024-11-21
> **Response to Reviewer JuSf**
>
> We genuinely thank the reviewer for leaning towards acceptance of our work. Below, we address the concerns in more detail.
>
> > (W.1 \& W.2) Table 2 lacks a setting of Baseline+TAR and classic metrics evaluation such as FVD.
>
> We sincerely thank reviewer JuSf for their constructive feedback and have included the Baseline + TAR results and added FVD evaluation, as shown below. Employing TAR alone can induce more dynamics, leading to improved MT-Score. Meanwhile, it achieves a lower FVD score compared to the baseline. This highlights the importance of TAR in synthesizing faithful long dynamic videos.
>
>
> |          | CLIP-VL | MT-Score | FVD |
> |----------|---------|----------|-----|
> | Baseline | 0.214   | 0.397    | 396 |
> | +TAR     | 0.219   | 0.409    | 310 |
>
>
> > (W.3)  When comparing VSTAR with FreeNoise in a mult-prompt setting, noticeable flickering in FreeNoise exhibits the implementation is not aligned with the original paper. (Although this does not affect my recognition of the effectiveness of VSTAR in the mult-prompt setting.)
>
> We appreciate the reviewer for their careful reading. We used the official implementation of FreeNoise, available [here](https://github.com/AILab-CVC/FreeNoise). We observed that it encounters challenges in the multi-prompt long-video setting, particularly when dynamic visual evolution is required. The flickering issue can also be observed from their provided examples, e.g., ["A boy/man/old man in grey clothes running in the summer/autumn/winter"](https://github.com/AILab-CVC/FreeNoise?tab=readme-ov-file#2-longer-multi-prompt-text-to-video-generation).
>
>
> > (Q.1) Concurrent work FreeLong
>
> We thank the reviewer for bringing this work to our attention. Indeed, FreeLong also notes a similar pattern in the temporal attention visualization. However, instead of directly regularizing temporal attention as VSTAR does, FreeLong performs transformations and high/low-pass filtering in the frequency domain. While this approach is insightful, it introduces challenges in determining appropriate parameters for the high/low-pass filters, making the process less intuitive. It is worth mentioning that VSTAR uses a single hyperparameter,  i.e., the Gaussian standard deviation $\sigma$ in the regulation matrix, to control the regularization strength and thus the video dynamics, without the need for frequency domain transformations, being simple yet effective. Additionally, our VSTAR can also transfer the dynamics from a given reference video, compared to FreeLong. Nevertheless, it would be interesting to further experiment and test the complementarity of both approaches when FreeLong is open-sourced. We once again thank the reviewer for their constructive feedback and the valuable pointer.

---

> ### Comment · Reviewer_JuSf · 2024-11-22
>
> Hi,
>
> First, I check the G.5 on your anonymous website again. Flickering also happens in VSTAR. There may be some errors on the web. Therefore I download the gif from the website. Then I find the frames are not well ordered in .gif. Could you check it and directly upload the original three videos in G.5 as a .zip file (through the OpenReview system)?
>
> Second, the official implementation of FreeNoise does not support the three prompts. In your implementation, the third prompt never happens, which is not aligned with the same demo shown by FreeNoise. Still, I will keep the positive score. However, if you are very confident about your implementation, you can upload the related code thus we can check it.

---

> > ### Author Response · Authors · 2024-11-22
> > **Response to Reviewer JuSf**
> >
> > We thank reviewer JuSf for this amazing catch! We have updated our anonymous website accordingly and attached the videos in G.5 as a supplementary zip file.
> >
> > We greatly appreciate the reviewer for pointing this out! Upon further examination, we found that indeed despite multi-prompts being passed as input, the FreeNoise implementation hardcoded to use only the first two prompts [here](https://github.com/AILab-CVC/FreeNoise/blob/faef8ed25000237a30673c74a633056812678246/scripts/evaluation/funcs.py#L172C1-L183C44). We have added a note about this in the revised manuscript.  Once again, we sincerely thank the reviewer for the insightful feedbacks!

---

### Official Review · Reviewer_dcFR · 2024-10-30

**Soundness:** 3
**Presentation:** 3
**Contribution:** 3
**Rating:** 6
**Confidence:** 5

**Summary:**

The paper proposes an attention regularization technique for long video generation. In addition to that, the paper also proposes to use existing LLMs to generate better video text descriptions. The main contribution of this work lies in the temporal attention regularization technique, which has been presented with good motivations and was able to regularize the temporal attention to mimic the ones in real videos. The other merit of this approach is that it can be directly applied to certain T2V models without retraining.

**Strengths:**

The main strength of this paper is the proposed Temporal Attention Regularization (TAR), which has been inspired by comparing temporal attention maps between the real and the generated videos. The motivation for this idea is clear and strong, and we can see the differences in attention maps. This helps motivate the use of the proposed TAR approach. In particular, the demonstration of attention to visual patterns offers convincing reasons why the regularization should be performed.

Another good thing about this TAR is that it could be applied to existing T2V models without re-training as long as the existing models satisfy certain conditions. The paper also performs extensive experiments to justify such a design, pertaining to the impact of resolutions and number of frames.

**Weaknesses:**

The per-layer temporal attention analysis part is not very clear in Sec. 3.2. Are the resolutions corresponding to the layer dimensions in the UNET? Does that occur in both the encoder and decoder parts of UNET?

The use of additional matrix "max" in Eqn. (3) needs to be further validated. The Toeplitz matrix should ensure that the values for distance frames will decrease. The multiplication of delta_A with the max function will only scale the values but does not change the ordering. I am just curious about the impact of this "max" function and how will affect the regularization performance. It appears that the paper does not offer a more detailed analysis of this part, including experimental validations.

The use of DreamSim in Sec. 4.1 could transfer the dynamics from real videos to the target one based on this delta_A. By looking at the results presented in Figure 8, it appears that the dynamics transfer also includes the style transfer. Intuitively, the attention map was used to regularize the interactions among frames to ensure that the evolving content was consistent with the real videos. With that being said, should we assume that these "dynamics" also model the video motions? If that is the case, the style transfer appears to be confusing here.  Is this more due to the use of DreamSim to calculate the similarity matrix? Please also clarify what are the captured dynamics by using the default delta_A.

Based on the discussions from Sec.5, there are limitations to applying the TAR to existing T2V models, depending on how the temporal attention was designed. Though this was not specifically mentioned, was this mainly due to the use of a Transformer backbone instead of the UNET? The Transformer does use frequently with the position embeddings. Most existing video diffusion models, however, are based on the Transformer architecture, such as W.A.L.T, Latte, and VDT. Please offer more comments in this regard by looking at these Transformer-based architecture and whether the proposed approach can be applied to them. Some of them are also open-sourced. The paper have already provided some analysis. But it will be appreciated with further analysis.

The use of VSP in Sec 4.3 is straightforward. After applying the LLM, does the new generated prompt considered as the whole prompt to be used in cross-condition? Or does the paper treat the generated prompts as sub-segments and sequentially input that into the model? Please also clarify that. Compared to StoryDiffusion, which also models long-range video generation, what will be the advantage of the VSP? In general, what is the VSP+TAR compared to StoryDiffusion in particular in terms of long-range video generation.

**Questions:**

There are a bunch of questions raised in the weakness section. I would appreciate it if the authors can address these questions, especially these related to the TAR.

---

> ### Author Response · Authors · 2024-11-21
> **Response to Reviewer dcFR (1/2)**
>
> We greatly appreciate the reviewer for taking the time to provide such a thorough and detailed evaluation, as well as their overall positive assessment. Below, we address the individual concerns in detail:
>
>
> > (Q.1) Clarification on per-layer temporal attention analysis, i.e, resolution and location in UNet.
>
> Following the common practice of attention visualization using mean averaging[1,2], we performed our visualization by averaging across all layers at different resolutions within the UNet.
>
>
> > (Q.2) "max" operation in Eq. (3)
>
> Our aim of using the max operation is to provide an automatic scaling mechanism in response to the Softmax operation in the original attention calculation. Otherwise, the values of the designed regularization matrix might not match the original attention values well, making it less effective or causing it to be overly applied. Despite the maximum being scaled based on the original attention, the regularization strength can be flexibly controlled by adjusting the $\sigma$ to control the relative attention among frames (i.e., the shape of the regularization matrix), thus the video dynamics. As shown in Fig. 9, reducing $\sigma$ results in stronger regularization, leading to more dynamic content.
>
>
>
> > (Q.3) By looking at the results presented in Figure 8, it appears that the dynamics transfer also includes the style transfer. Intuitively, the attention map was used to regularize the interactions among frames to ensure that the evolving content was consistent with the real videos. With that being said, should we assume that these "dynamics" also model the video motions? If that is the case, the style transfer appears to be confusing here. Please also clarify what are the captured dynamics by using the default $\Delta_A$.
>
> We thank the reviewer for this thoughtful observation. Based on our extensive experiments, we did not observe that the dynamics transfer includes a style transfer effect. To further illustrate this, we have provided more samples using different random seeds [here](https://anonymous.4open.science/r/VSTAR-ICLR-Rebuttal/Reviewer_dcFR/README.md). It can be seen that the scenes look quite different, and the third sample even contains trees and a lake.
>
> From a technical point of view, the proposed regularization is applied to temporal attention, which only models the correlations between different frames and doesn't affect the style of the generated frames. Our designed $\Delta_A$ promotes stronger correlations with neighboring frames, while correlations decrease for more distant frames. This mimics the natural behavior of real videos with uniform visual progression, without special visual effects such as Flash Zoom or Hyperzoom.
>
>
>  > (Q.4) Based on the discussions from Sec.5, there are limitations to applying the TAR to existing T2V models, depending on how the temporal attention was designed. Please offer more comments on Transformer-based architecture, which frequently uses positional encodings.
>
> We thank the reviewer for the insightful comments. Indeed, transformer-based architectures often rely on positional encoding, which largely constrain their generalization ability for longer video synthesis. Similar observation has been identified for LLMs, as mentioned in L523. Furthermore,  recent T2V models such as Open-Sora-Plan and CogVideoX even utilize 3D full attention without explicitly modeling the temporal dimension, rendering VSTAR inapplicable in this case. Nevertheless, VSTAR can be used to improve other T2V models, e.g., VideoCrafter and ModelScope. We once again thank reviewer dcFR for their constructive feedback and have extended our discussion in the revised manuscript.
>
> [1] Hertz, Amir, et al. "Prompt-to-Prompt Image Editing with Cross-Attention Control." ICLR 2023
>
> [2] Chefer, Hila, et al. "Attend-and-excite: Attention-based semantic guidance for text-to-image diffusion models." ACM Transactions on Graphics (TOG) 42.4 (2023): 1-10.

---

> > ### Author Response · Authors · 2024-11-21
> > **Response to Reviewer dcFR (2/2)**
> >
> > > (Q.5.1) The use of VSP in Sec 4.3 is straightforward. Is the new generated prompt considered as the whole prompt or as sub-segments and sequentially input that into the model?
> >
> > We first obtain textual embeddings based on the subprompts of several key frames. These embeddings are then interpolated to produce a unique textual embedding for each individual frame. These embeddings are utilized by the model via cross-attention to guide the synthesis process, in a single inference pass. We acknowledge that the original illustration may not have been sufficiently clear and sincerely thank the reviewer for pointing this out. We have updated Fig.2 and text in Sec. 3.4 for improved clarity and included additional details in the Sec. C of the Supplementary Material.
> >
> >
> > > (Q.5.2) Comparison with StoryDiffusion.
> >
> > We thank the reviewer dcFR for bringing this work to our attention. Unlike our VSTAR, which generates videos directly in a single pass, StoryDiffusion first generates subject-consistent images, then inserts frames in between to form a video. This is more similar to FreeBloom, facing consistency challenges for longer video generation, compared to direct video synthesis, as shown in Table 1 and G.5 in the Supplementary GitHub Link [here](https://anonymous.4open.science/r/VSTAR).
> > It is worth noting that StoryDiffusion specifically targets subject-consistent settings. VSTAR has more general usage, can also generate time-elapsed videos with more visual appearance changes, e.g., “A landscape transitioning from winter to spring” in Fig. 7.
> >
> > We hope our detailed responses sufficiently address the reviewer’s feedback, and we are happy to provide further clarifications if there are any remaining questions.

---

> > > ### Comment · Reviewer_dcFR · 2024-11-28
> > >
> > > Thanks for your replies. Most of my concerns are addressed. The proposed TAR is interesting and does show its effectiveness. However, considering that most of the SOTA T2V models are based on the Transformer, coupled with positional encoding, the TAR shows certain limitations in its practical usage. Therefore, I still maintain my original score.

---

### Official Review · Reviewer_4ewj · 2024-10-31

**Soundness:** 3
**Presentation:** 3
**Contribution:** 3
**Rating:** 6
**Confidence:** 5

**Summary:**

The authors systematically analyze the role of temporal attention in Video Generation and contribute a training-free technique called “Generative Temporal Nursing” to create more temporal dynamics for long video generation.  Besides, LLM-based video synopsis prompting is explored to facilitate gradual visual changes. Experimental results show the effectiveness of the proposed methods in terms of enhancing video dynamics.

**Strengths:**

1. The idea is simple and easy to follow.
2. The motivation of the method is reasonable and strong.
3. The analysis of temporal attention in T2V model may benefit other research.
4. The paper is well-written and well-organized.

**Weaknesses:**

1. The introduction to the VSP is short and some of the details are not clear: do all frames share one interpolated text embedding or does each group share a different embedding?
2. Introducing a Toeplitz matrix to temporal attention could help to improve temporal dynamics. What I am concerned about is that this kind of hard modification may break the original motion, since I find that the motions of Superman and Spiderman are wired.
3. I wonder about the proportion of the "less structured" temporal attention maps in video diffusion models because only some cases and layers do not satisfy the band-matrix pattern according to my personal experience.
4. It seems that MTScore from ChronoMagic-Bench is adopted to evaluate the video dynamics. I want to see other metrics in the same benchmark like Coherence Score (CHScore). I am happy to raise my rate if more quantitative results are provided.

**Questions:**

Please refer to the weakness part.

---

> ### Author Response · Authors · 2024-11-21
> **Response to Reviewer 4ewj**
>
> We thank the reviewer for the overall positive assessment. In what follows, we address the individual concerns in detail:
>
>
> > (W.1) Details on VSP: Do all frames share one interpolated text embedding or does each group share a different embedding?
>
> To clarify, each frame has a unique textual embedding, obtained by interpolating between the textual embeddings of key frames. These embeddings are then utilized by the model via cross-attention mechanism to guide the synthesis process. To illustrate this, we have visualized the interpolation process through color coding in Fig. 2. We thank the reviewer for pointing this out, and we have updated Fig.2 and text in Sec. 3.4 for improved clarity and also included these details in the Sec. C of the Supplementary Material.
>
>
> > (W.2) Introducing a Toeplitz matrix to temporal attention could help to improve temporal dynamics. There is concern that hard temporal modification may break the original motion.
>
> It is a very good question, which we also asked ourselves when we were designing the solution. We agree that, in theory, extreme regularization could potentially disrupt the original motion.
> Thus we carefully designed the regularization matrix as a Toeplitz matrix, with values along the off-diagonal direction following a Gaussian distribution to explicitly avoid abrupt change.
> As a result, in our experiments we did not observe evident abrupt changes in motion. Also, as shown in Fig. 9, users have the flexibility to control the strength of the regularization by adjusting the standard deviation ($\sigma$) of the Gaussian, which we have visualized in Fig. S.13. This allows users to customize the extent of visual changes based on their preferences. As mentioned in L485, we find that $\sigma_{64}=1$ strikes a good balance between dynamic changes and temporal coherency.
>
> Meanwhile, we agree with the reviewer that sometimes the motion may look less natural, which is an inherited issue from the pretrained T2V models, as VSTAR is training-free. Nevertheless, we are optimistic that the community will open-source better and larger models trained on more data, further enhancing the motion quality. We thank reviewer 4ewj for the constructive feedback, and we have extended our limitation discussion in Sec. E of the Supplementary Material to address this.
>
>
> > (W.3) The proportion of the "less structured" temporal attention maps in video diffusion models, based on my personal experience only some cases and layers do not satisfy the band-matrix pattern.
>
> We thank the reviewer for sharing their experiences. They are quite align with ours on some models, e.g., LaVie and AnimateDiff, exhibit more structured temporal attention for shorter videos (e.g., 16 frames). However, their attention maps degrade significantly for longer videos, such as those with 32 frames, as illustrated in Fig. 10.  This phenomenon was commonly observed across layers and was easily noticeable when generating longer videos.
> We attribute this to their positional encoding of the frame index, which restricts their ability to generalize effectively for longer video synthesis beyond the trained number of frames, e.g., 16 frames, as discussed in Sec. 5.
>
>
> > (W.4) Other metrics like CHScore
>
> We appreciate reviewer 4ewj's pointer and added the CHScore evaluation below. It can be seen that VSTAR achieves the best vision-language alignment and MTScore, while achieving competitive CHScore. For reference, we also computed the CHScore of dynamic videos, yielding 10.96, which is very close to the scores achieved by VSTAR. We have included some CHScore for real and generated videos [here](https://anonymous.4open.science/r/VSTAR-ICLR-Rebuttal/Reviewer_4ewj/README.md).
>
>
> |              | LaVie | V.Crafter2 | CogVideoX | FreeBloom | FreeNoise | VSTAR     |
> |--------------|-------|------------|-----------|-----------|-----------|-----------|
> | VSP          | ✔︎     | ✔︎          | ✗         | ✔︎         | ✔︎         | ✔︎         |
> | Single-Pass  | ✔︎     | ✔︎          | ✔︎         | ✗         | ✗         | ✔︎         |
> | CLIP-VL⭡    | 0.221 | 0.220      | 0.221     | 0.219     | 0.224     | **0.233** |
> | MT-Score⭡   | 0.306 |  0.401     | 0.312     | 0.361     | 0.379     | **0.448** |
> | CH-Score⭡   | 9.70  | 11.02      | **11.64** | 4.21      | 6.83      | 10.87     |
>
> We hope our responses have addressed the reviewer’s concerns, and we are happy to provide further clarifications if there are any additional questions.

---

> > ### Comment · Reviewer_4ewj · 2024-11-26
> >
> > Thank you for your response and the additional results.  Most of my concerns have been addressed. This is a good work. However, it does have limitations that I care about. After thorough consideration, I believe the previous score sufficiently represents the quality of the article, so I decided to maintain the current score.

---

### Official Review · Reviewer_Apsy · 2024-11-04

**Soundness:** 2
**Presentation:** 3
**Contribution:** 2
**Rating:** 6
**Confidence:** 4

**Summary:**

The paper proposes a new concept of "Generative Temporal Nursing" and aims at enhancing the temporal dynamics of long video synthesis without requiring additional training or introducing significant computational overhead during inference. Specifically, this paper first find out that the temporal correlation varies a lot betwen frames, and introduce TAR module to adjust the temporal correlation with regularization. Then this paper propose VSP module to provide key visual states to decompose the transition along the temporal axis with LLM.

**Strengths:**

1. The analysis of temporal correlation is meaningful and interesting.

2. TAR module is well-motivated, effective and readily applicable to pre-trained T2V models.

3. The paper is well-structured and clearly presents the methodology, experiments, and results.

**Weaknesses:**

1. The side effect of TAR: In the Fig.5, the introduction of TAR in generation models may reduce the amplitude of motion.

2. Further ablation on TAR. Does VSP module is used in Fig.9? How do the TAR module and attention map change with the introduction of VSP module?

3. Temporal consistency. Does the introduction of VSP lead to a decrease in temporal consistency? Can authors provide some video demos?

**Questions:**

Please see the weakness.

---

> ### Author Response · Authors · 2024-11-21
> **Response to Reviewer Apsy**
>
> We sincerely thank the reviewer for the positive assessment of our work. In what follows, we address the individual concerns in detail:
>
> > (W.1) The side effect of TAR: may reduce motion amplitude.
>
> We thank reviewer Apsy for the engaging question. In our experiments, we didn't observe TAR to reduce the motion amplitude. Instead, we saw that TAR enables synthesis of more dynamic content. For instance, from more visual examples provided in G.3 of the  [supplementary anonymous Github link](https://anonymous.4open.science/r/VSTAR), we can see that the corgi in VSTAR's result has a more dynamic walk. To further consolidate our observations, we conducted a quantitative evaluation using MTScore - a new metric specifically designed to measure metamorphic amplitude, the results can be found in Table 1. Using TAR improves MTScore from 0.401 to 0.448. We also added samples with and without using TAR as well as their individual MTScores [here](https://anonymous.4open.science/r/VSTAR-ICLR-Rebuttal/Reviewer_Apsy/README.md). Our approach demonstrates superior performance compared to other methods, achieving a higher MTScore, which reflects more pronounced and dynamic transformations over time. These results highlight VSTAR’s capability to generate long videos with dynamic content.
>
>
> > (W.2) Further ablation on TAR. Does VSP module is used in Fig.9? How do the TAR module and attention map change with the introduction of VSP module?
>
> To clarify, the VSP module is indeed utilized in the results presented in Fig. 9. To further analyze the role of VSP, we have provided the attention map visualization with VSP only [here](https://anonymous.4open.science/r/VSTAR-ICLR-Rebuttal/Reviewer_Apsy/README.md). These visualizations demonstrate that using VSP alone does little to improve the temporal attention structure, and the video output shows minimal visual evolution, even when the prompts describe visual changes. In contrast, when combined with TAR, the temporal attention becomes more structured, and the results align with the desired content. This highlights the crucial role of TAR in generating long dynamic videos. We have incorporated these discussions in  Sec. A.5 of the manuscript accordingly.
>
>
> > (W.3) Temporal consistency. Does the introduction of VSP lead to a decrease in temporal consistency?
>
> We thank the reviewer for raising this insightful question. VSP's impact on temporal consistency can vary depending on the prompts used. When the prompts describe numerous abrupt or significant changes over a relatively short video, it may compromise temporal consistency. For instance, as shown in the example [here](https://anonymous.4open.science/r/VSTAR-ICLR-Rebuttal/Reviewer_Apsy/README.md), when the described subject remains the same (e.g., A Ferrari) while undergoing progressive seasonal changes, the result exhibits good consistency. However, when the prompt instruct the subject to change completely (e.g., switching from a Ferrari to a truck or limousine), as expected, we observe a degradation in temporal consistency. To generate more reasonable subprompts, as described in Sec. C, L1182 of the Supplementary Material, we explicitly instruct the LLMs to consider coherence between subprompts. Additionally, we interpolate between the textual embeddings of several key prompts, rather than generating prompts for each frame individually. This ensures smooth transitions in the textual embeddings, avoiding abrupt changes. Nevertheless, when the described visual changes are too extensive, it may create an impression of reduced consistency.
>
>
> We hope our responses have addressed the reviewer’s concerns, and we are happy to provide further clarifications if there are any remaining questions.

---

> > ### Comment · Reviewer_Apsy · 2024-11-27
> >
> > Thanks for your response, which has addressed most of my concerns.  I decide to keep my rating.

---

### Author Response · Authors · 2024-12-02
**Thank You for the Positive Assessment and Feedback**

Dear Reviewers,

We sincerely appreciate that all reviewers provided a positive assessment of our work and leaned toward acceptance. Your constructive suggestions and thoughtful engagement have been invaluable in improving our manuscript. We are pleased that most concerns from the initial reviews have been addressed through the rebuttal.

We are especially encouraged by your recognition of the key contributions and strengths of our paper, including:

- The proposed VSTAR method effectively enables pre-trained T2V models to generate longer dynamic videos in a single inference pass. (Apsy, dcFR, JuSf)
- The analysis of temporal attention in T2V models, which provides inspiring insights beneficial to future research. (Apsy, 4ewj, dcFR, JuSf)
- The paper is well-structured and well-written. (Apsy, 4ewj)

Thank you once again for your thoughtful feedback and for acknowledging the potential impact of our work!

---

### Meta-Review · Area_Chair_Eqkb · 2024-12-17

**Metareview:**

The paper introduces a training-free method for generating longer dynamic videos. Two contributions were claimed, the main one being a technique to regularize the temporal attention using a Toeplitz matrix, and another contribution to generate more enriched text prompts, which enable improved temporal dynamics and coherent video generation. The method's strength lies in its applicability to pre-trained models without fine-tuning. While concerns about potential motion amplitude reduction and applicability to Transformer-based models remain, the authors provided convincing rebuttals. Overall, the consensus leans towards acceptance, recognizing the paper’s contributions to enhancing long video synthesis.

**Additional Comments On Reviewer Discussion:**

Reviewers raised concerns about motion amplitude, temporal consistency, and TAR’s applicability to Transformers. Authors clarified TAR's impact, provided additional metrics (FVD, CHScore), and addressed VSP details. They demonstrated TAR’s effectiveness and discussed limitations with Transformer-based models. The rebuttals were thorough, addressing key concerns. Despite minor limitations, the method’s novelty, training-free approach, and clear improvements in video dynamics justified acceptance. The reviewers maintained positive ratings, and the overall consensus supports the paper’s contributions.

---

### Decision · Program_Chairs · 2025-01-22

Accept (Poster)